# Cluster-type analogue memristor by engineering redox dynamics for high-performance neuromorphic computing

Jaehyun Kang [1,2], Taeyoon Kim [1], Suman Hu[1], Jaewook Kim[1], Joon Young Kwak [1], Jongkil Park[1], Jong Keuk Park[1], Inho Kim[1], Suyoun Lee [1], Sangbum Kim [2] & YeonJoo Jeong [1✉]

Memristors, or memristive devices, have attracted tremendous interest in neuromorphic hardware implementation. However, the high electric-field dependence in conventional filamentary memristors results in either digital-like conductance updates or gradual switching only in a limited dynamic range. Here, we address the switching parameter, the reduction probability of Ag cations in the switching medium, and ultimately demonstrate a cluster-type analogue memristor. Ti nanoclusters are embedded into densified amorphous Si for the following reasons: low standard reduction potential, thermodynamic miscibility with Si, and alloy formation with Ag. These Ti clusters effectively induce the electrochemical reduction activity of Ag cations and allow linear potentiation/depression in tandem with a large conductance range (~244) and long data retention (~99% at 1 hour). Moreover, according to the reduction potentials of incorporated metals (Pt, Ta, W, and Ti), the extent of linearity improvement is selectively tuneable. Image processing simulation proves that the $Ti_{4.8\%}$:a-Si device can fully function with high accuracy as an ideal synaptic model.

[1] Center for Neuromorphic Engineering, Korea Institute of Science and Technology, Seoul 02792, Republic of Korea. [2] Department of Materials Science and Engineering, Seoul National University, Seoul 08826, Republic of Korea. ✉email: jeongyeonjoo@kist.re.kr

After the experimental demonstration of the two-terminal metal-insulator-metal (MIM) structure memristive system in 2008[1], extensive research on nanoscale resistive switching devices has been conducted in diverse fields of application, from digital-based logic and memory[2–5] to analogue-based artificial synaptic elements for neuromorphic computing[6–10]. The crossbar-structured memristors or memristive devices (throughout the paper, we use the term "memristor" to refer to a memristive device in short) in Fig. 1a exhibited prospective capabilities in computing tasks, such as signal processing and image recognition with experimental demonstrations[4,11,12]. Conductive-bridge random access memory (CBRAM) is a type of memristor which utilises the redox and ion migration process of active metals to form or rupture conductive filaments (CFs), resulting in resistance changes[2,8,13]. The engineering of various materials and structures of CBRAM has been attempted to

improve its characteristics after the early demonstration of analogue synaptic behaviours[10]; switching performance variation was minimised by the one-dimensional CF confinement effect[9], sub-femtojoule power consumption was achieved by the formation of atomically thin CF[14], and retention time was improved through filament interfacial energy stabilisation[15]. However, limited improvements have been made for analogue linearity and dynamic range, only showing on/off ratios typically less than 10 in the analogue switching region[16]. Few studies achieved high on/off ratios in analogue CBRAM by applying large voltage pulses, yet the linearity significantly deteriorated to almost digital-like switching[9,17–19]. This trade-off between linearity and on/off ratio strongly originates from the positive feedback effect in an electric field during the filament growth process, where the electric field induces exponential ionic migration toward the depleted filament region[2,13,20–22]. A memristor

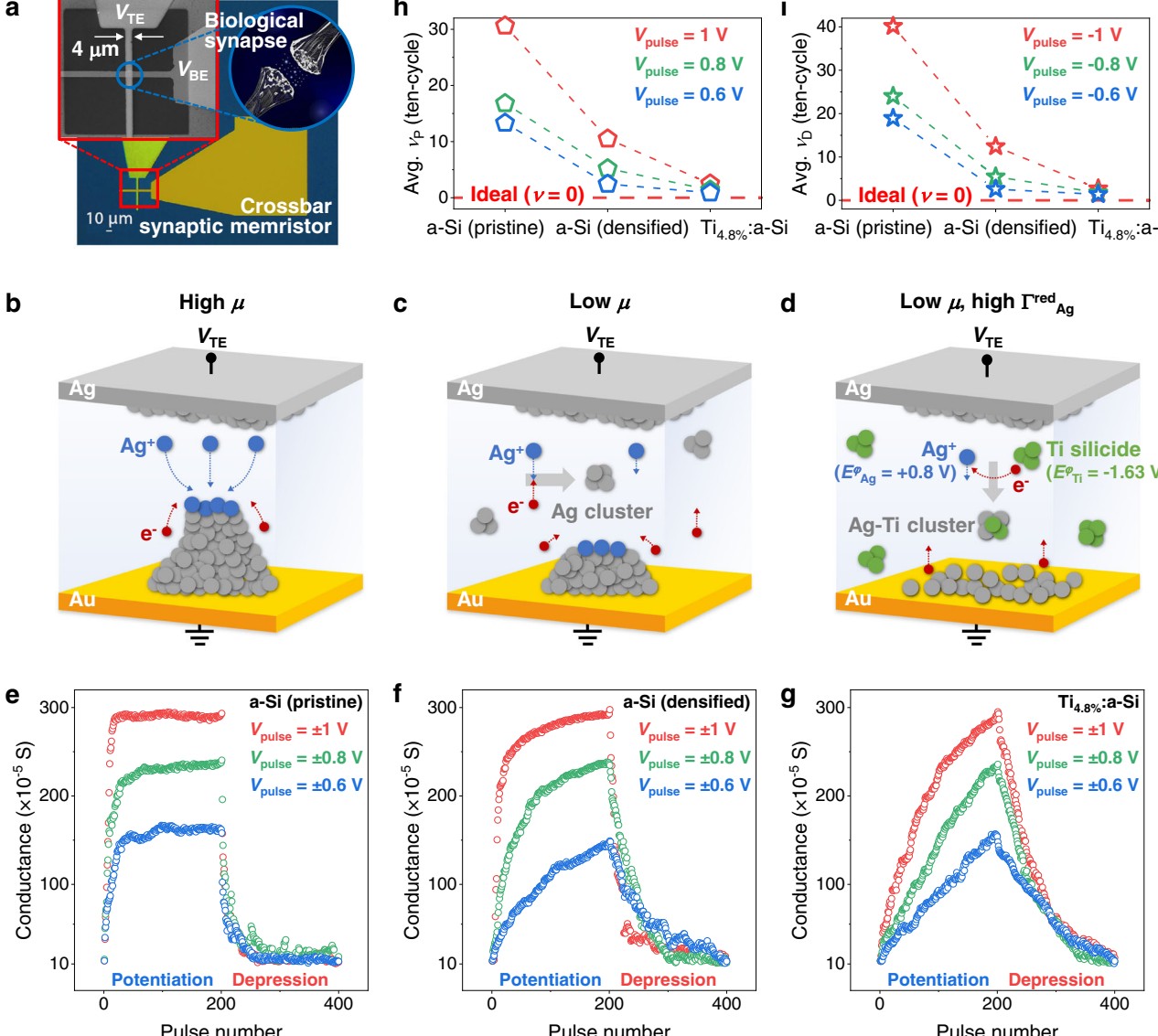

**Fig. 1 The effect of cluster-type switching dynamics in analogue linearity. a** Microscopic image of an a-Si synaptic memristor (left inset: scanning electron microscopy (SEM) image of crossbar structure, right inset: illustrated biological synapse). Device schematics with operation mechanism illustration for each memristor situation, **b** high cation mobility ($\mu$) case for a-Si (pristine) device, **c** low $\mu$ case for a-Si (densified) device, and **d** low $\mu$ and high Ag reduction probability ($\Gamma^{red}_{Ag}$) case for Ti$_{4.8\%}$:a-Si device. Grey, blue, green, and red spheres represent Ag atoms, Ag ions, Ti atoms, and electrons, respectively. $E^{\varphi}_{Ag}$ and $E^{\varphi}_{Ti}$ represent standard reduction potentials of Ag and Ti. Analogue conductance updates under three different pulse conditions (blue: 0.6/−0.6 V, green: 0.8/−0.8 V, red: 1/−1 V with 1 μs duration for potentiation/depression) in **e** a-Si (pristine), **f** a-Si (densified), **g** Ti$_{4.8\%}$:a-Si devices. The conductance was measured with a read pulse (0.1 V, 1 μs) after each programming pulse. **h** Average potentiation and **i** depression nonlinearity factors of a-Si (pristine), a-Si (densified), and Ti$_{4.8\%}$:a-Si memristors for ten-cycle at each pulse condition.

with a linear conductance update and a large on/off ratio is highly challenging to realise and yet to be developed but is desired for superior performance in neuromorphic computing[4,6,16].

In this work, we propose a cluster-type analogue CBRAM in which the switching is dependent on the amount of Ag-clusters instead of filament formation, expecting a significant reduction of feedback effect in an electric field. The essence of the study is to control the redox dynamics of active metals, rendering Ag cations to be reduced inside the amorphous Si (a-Si) switching medium rather than in the a-Si/counter electrode interface. We first densely packed a-Si to limit the ionic diffusion pathways and further conceived a unique reducing agent concept to enlarge the Ag reduction probability in the a-Si layer. This goal was achieved without additional structural complexity by incorporating transition-metal (TM) nanoclusters inside the Si medium. Ti was chosen for satisfying the following three requirements: lower standard reduction potential ($E^\varphi$) than Ag to favourably give away its electrons to Ag cations, the presence of TM-silicide to prevent TM-based resistive switching, and alloy formation with Ag. The densified $Ti_{4.8\%}$:a-Si CBRAM exhibited near-ideal linearity, increased on/off ratio, and long data retention. The nucleated Ag particles in the $Ti_{4.8\%}$:a-Si layer were visualised through field-emission transmission electron microscopy (TEM). The proposed mechanism was further verified by examining various metals having different $E^\varphi$, such as Pt, W, and Ta. Finally, we simulated two image processing tasks based on our $Ti_{4.8\%}$:a-Si device characteristics to demonstrate comparable data processing ability to the ideal case.

## Results

### Cluster-type memristor by engineering reduction probability.

It is widely known that packing densities of amorphous switching matrix critically influence the cationic transport process of CBRAM[2,5,23]; low-density amorphous films demonstrate high cation mobility ($\mu$), where the injected cations reach the counter electrode in a short time and reduce at the surface of the counter electrode[2,3,5,23,24]. This typical filament formation mechanism can be modulated in densely packed amorphous materials with lower $\mu$[23,25]. Cations can recombine inside the dielectric at a higher probability by capturing the free electrons from the cathode, building metal clusters rather than filaments. The cluster-type may improve the analogue linearity due to its smaller feedback effect in electric-field than in the filament-based CBRAM. Based on these postulations, we fabricated two different microstructures of a-Si CBRAM with Ag as an active metal (Fig. 1b, c) to verify the hypothesis. Considering the wide amorphous density range of a-Si ($1.75–2.2\ g\ cm^{-3}$)[26,27], we fabricated pristine and densified 7-nm-thick a-Si CBRAM. First, the a-Si (pristine) device showed 16.3% less density ($1.95\ g\ cm^{-3}$, Supplementary Fig. 3) than crystal silicon density ($2.33\ g\ cm^{-3}$)[26,27] due to disordered micro-voids and grain boundaries which can offer fast ion migration pathways with high $\mu$. Hence, the conventional filament growth dynamics apply to the a-Si (pristine) device as depicted in Fig. 1b, where the CF grows from the counter electrode and inherently possesses strong positive feedback in the local electric field[13,20,21], resulting in abrupt resistive switching. We speculate that this filament-based switching process critically deteriorates linear switching properties. To characterise the analogue switching of the a-Si (pristine) device, we applied three different sets of 200 potentiation/depression (P/D) identical programming pulse trains (0.6/−0.6, 0.8/−0.8, and 1/−1 V, 1 μs) to the top electrode ($V_{TE}$). As expected, the analogue performance of the a-Si (pristine) device displayed a highly nonlinear conductance update even at the lowest pulse bias condition (Fig. 1e), and the trade-off relationship between linearity and on/off ratio was clearly marked. As pulse amplitude changed from 0.6/−0.6 to 1/−1 V, the dynamic range was enhanced, whereas the linearity indicator ($\nu$) for each P ($\nu_P$) and D ($\nu_D$)

averaging from the ten-cycle measurement degraded, as summarised in Fig. 1h, i (see Methods for the $\nu$ calculation and Supplementary Fig. 4 for ten-P/D-cycle). Therefore, through simple pulse engineering, it is difficult to achieve both the linear analogue update and a large dynamic range simultaneously in a filament-based memristor. In contrast, the a-Si (densified) memristor was fabricated with only 3.0% less density ($2.26\ g\ cm^{-3}$, Supplementary Fig. 3) than c-Si. We speculate that the densely packed Si clusters with slow ion migration ensure Ag cations to possess a more temporal margin to capture incoming free electrons from the cathode, and accordingly, the a-Si (densified) device acts for a low $\mu$ (Fig. 1c) situation, which creates Ag-clusters in the a-Si layer. At the same pulse condition, the a-Si (densified) device showed considerably linear switching (Fig. 1f), with a 65.7% decreased average $\nu_P$ than a-Si (pristine) device at 1/−1 V pulse (Fig. 1h, i). The limited ionic transport channel in the a-Si (densified) device effectively lessened the accumulation of Ag in the counter electrode through its recombination inside the Si medium. This partially relieved the positive feedback effect that influenced the nonlinear analogue switching.

Despite the enhanced analogue linearity performance, the a-Si (densified) device still suffered from nonlinear conductance update issues at high pulse amplitude, which is essential to achieving a high dynamic range. To further control and enlarge the Ag cation reduction capability inside the Si matrix, we suggest an approach that regulates the reduction probability of cations during the switching process. We examined TM elements that possess low (negative) $E^\varphi$ to enable the transfer of electrons from TM-clusters to Ag cations. Ag cations can be possibly reduced through interfacial charge-transfer processes mediated by the TM reducing agents[28,29]. Here, we integrated Ti atoms in a densified a-Si layer for its much lower $E^\varphi$ value ($E^\varphi_{Ti} = -1.63\ V$) than Ag ($E^\varphi_{Ag} = +0.8\ V$), which can capture and reduce migrating Ag cations to Ag-clusters (The $E^\varphi$ values used in this paper were taken from ref. [30]). This large reduction potential difference—$\Delta E^\varphi_{Ag\text{-}Ti}$ value of 2.43 V—effectively increases the reduction probability of Ag cations in a-Si medium, and accordingly, the $Ti_{4.8\%}$:a-Si device in Fig. 1d represents a low $\mu$ and high Ag reduction probability ($\Gamma^{red}_{Ag}$) situation. We finely tuned the Ti amount in a densified a-Si layer so as not to affect the initial resistance state of the device (see Methods for details), and the Ti ratio of 4.8% in a-Si film was measured by X-ray photoelectron spectroscopy (XPS) measurements. As shown in Fig. 1g, the $Ti_{4.8\%}$:a-Si device exhibited an almost ideally linear switching with 0.89 average $\nu_P$ at 0.6/−0.6 V pulse condition (Fig. 1h, i). To confirm that the resistive switching was solely driven by Ag ion migration, not by the incorporated Ti elements, $Ti_{4.8\%}$:a-Si device without an Ag layer in the top electrode was tested by multiple quasi-static current-voltage ($I$-$V$) sweeps, where the resistive switching was not observed (Supplementary Fig. 5). We believe that due to the thermodynamically stable Ti-Si cluster[31,32], the formation of silicide solely assures Ag-based analogue switching. It is notable that in the aforementioned a-Si (densified) device, we utilised less controllable free electrons injected from the cathode for the Ag cation reduction. In contrast, the new approach uses electrons of Ti, and the recombination probability can be modulated with higher flexibility by changing the amount of Ti or other TMs possessing different $E^\varphi$. These mechanisms of cluster-type analogue switching certainly lessened the positive feedback effect of an electric field and resolved the chronic trade-off problem (linearity and on/off ratio) of most filament-based CBRAMs, as shown in Supplementary Fig. 6, maintaining an exceptionally low nonlinearity factor at all pulse amplitudes.

### Advanced analogue characteristics and direct microstructure analysis of the $Ti_{4.8\%}$:a-Si memristor.

Accomplishing a high on/off ratio is another important factor to enable as many

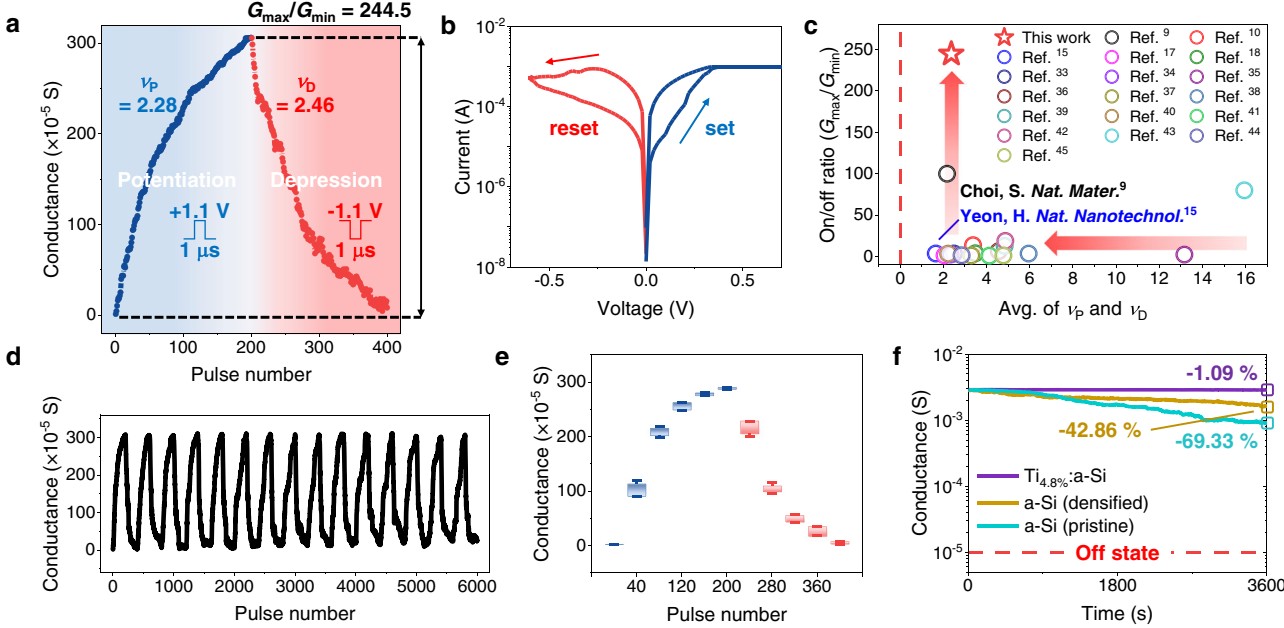

**Fig. 2 The electrical characteristics of the Ti$_{4.8\%}$:a-Si memristor. a** Linear conductance update in the full analogue range under the 1.1/−1.1 V, 1 μs pulse condition with 0.1 V, and 1 μs read pulse. **b** The observed quasi-static current-voltage (*I-V*) switching characteristics after forming process (forming voltage of ~1.5 V) with a compliance current of 1 mA. **c** Benchmark of the Ti$_{4.8\%}$:a-Si memristor with previously reported Ag[9,10,15,17,18,33–43]- and Cu[44,45]-based conductive-bridge random access memory (CBRAM) devices for linearity and on/off ratio characteristics. **d** Endurance test of 15-cycle conductance updates performed under the 1.1/−1.1 V, 1 μs pulse condition. **e** Box plot of ten-cycle under the 1.1/−1.1 V, 1 μs pulse condition, extracting intermediate conductance value at specific pulse number. The box portion is defined by the 25[th] and 75[th] percentile, with whisker length set to mean ± s.d. **f** Improved analogue data retention capability of the Ti$_{4.8\%}$:a-Si device compared to the a-Si (pristine) and a-Si (densified) devices, measured by 0.1 V read bias. The devices were programmed to maximum conductance state (G$_{max}$) under the 1/−1 V, 1 μs pulse condition before the retention test.

distinguishable conductance levels as with a practical sensing circuit and, at the same time, to secure improved noise margin of a neuromorphic system[4,6,16]. In Fig. 1e–g, all the analogue measurements were done by fixing the minimum conductance levels (G$_{min}$) to 0.1 mS to ensure a fair comparison in different devices, and therefore, the Ti$_{4.8\%}$:a-Si device only showed a part of its available analogue dynamic range. As shown in Fig. 2a, the Ti$_{4.8\%}$:a-Si device realised a tremendously high analogue on/off ratio of 244 (1.1/−1.1 V, 1 μs), along with a linear conductance update (ν$_P$ = 2.28 and ν$_D$ = 2.46). Therefore, the conductance can be gradually updated by more than two orders of magnitude, and the cluster-type switching dynamics are believed to be maintained throughout the wide conductance range in the Ti$_{4.8\%}$:a-Si device. The resistive switching characteristic of the Ti$_{4.8\%}$:a-Si device was also observed in the quasi-static *I-V* curve after the forming process (Fig. 2b). Interestingly, the *I-V* set/reset processes in the Ti$_{4.8\%}$:a-Si device contrasted with much more abrupt *I-V* set/reset processes occurring in the a-Si (pristine) device (Supplementary Fig. 1a). To our best knowledge, the two main features of analogue devices—the linear switching performance and the large dynamic range of Ti$_{4.8\%}$:a-Si device—display a clear deviation from the previously reported Ag[9,10,15,17,18,33–43]- and Cu[44,45]-based analogue CBRAM (Fig. 2c). These outstanding analogue properties prove the remarkable effectiveness of the proposed cluster-type analogue switching.

Endurance, variation, and retention are the key measurements for the reliability of a synaptic device that have been studied in various analogue resistive switching devices[4,6,16,46]. As shown in Fig. 2d, the Ti$_{4.8\%}$:a-Si device was tested through a total of 6000 programming pulses of repeated 200 P/D. We also present ten-cycle P/D switching as a box plot, extracting intermediate conductance states at specific pulse numbers (Fig. 2e) and device-to-device variation along with standard deviation (s.d.) at each P/D pulse number (Supplementary Fig. 7). In terms of retention performance, our Ti-assisted device

demonstrated substantially long-term data stability at a maximum conductance state (G$_{max}$) after potentiation (Fig. 2f). Owing to thermodynamic immiscibility between Ag and Si[47], Ag filament stochastically dissolved in a-Si (pristine) matrix, and after an hour of retention test, the device showed 69.3% conductance decay from its initial state. Given the situation of the a-Si (densified) device, the improved data retention of a 42.86% decay was attributed to the lower μ that slowed down the Ag filament dissolution process. Regarding the Ag-Ti phase diagram[48], we anticipated Ti-clusters to easily form an Ag-Ti alloy and stabilise the high interfacial energy of Ag in the Si matrix. Indeed, as shown in Fig. 2f, Ti$_{4.8\%}$:a-Si device outstood in the data retention performance with only 1.1% conductance decay after an hour at room temperature. More importantly, the Ti$_{4.8\%}$:a-Si device sustained considerably low conductance decay percentage at multi-conductance levels at room temperature and demonstrated stable retention performance with elevated temperatures (100 °C, 150 °C, and 200 °C) at low-, mid-, and high-conductance levels (Supplementary Fig. 8). We conclude that exploiting the thermodynamic miscibility between active metal and incorporated reducing agent is desirable to extend the stability of stored analogue data[15]. Therefore, we have devised a cluster-type CBRAM that exhibits superior analogue performance in linearity, dynamic range, and data retention. Our findings can guide the way for the further development of analogue synaptic memristors.

To directly observe the Ag nucleation inside the a-Si layer, we performed a bright-field TEM imaging before/after programming the Ti$_{4.8\%}$:a-Si devices. The TEM specimens were prepared by focused-ion-beam (FIB) lift-out and thinning processes. As shown in Fig. 3a, TEM images taken after programming successfully revealed our hypothesis on the Ag-cluster nucleation inside the dielectric layer in the Ti$_{4.8\%}$:a-Si device. The lattice fringes in the fast Fourier transform, which results from the high-resolution TEM image of Ag-cluster in the a-Si layer could be

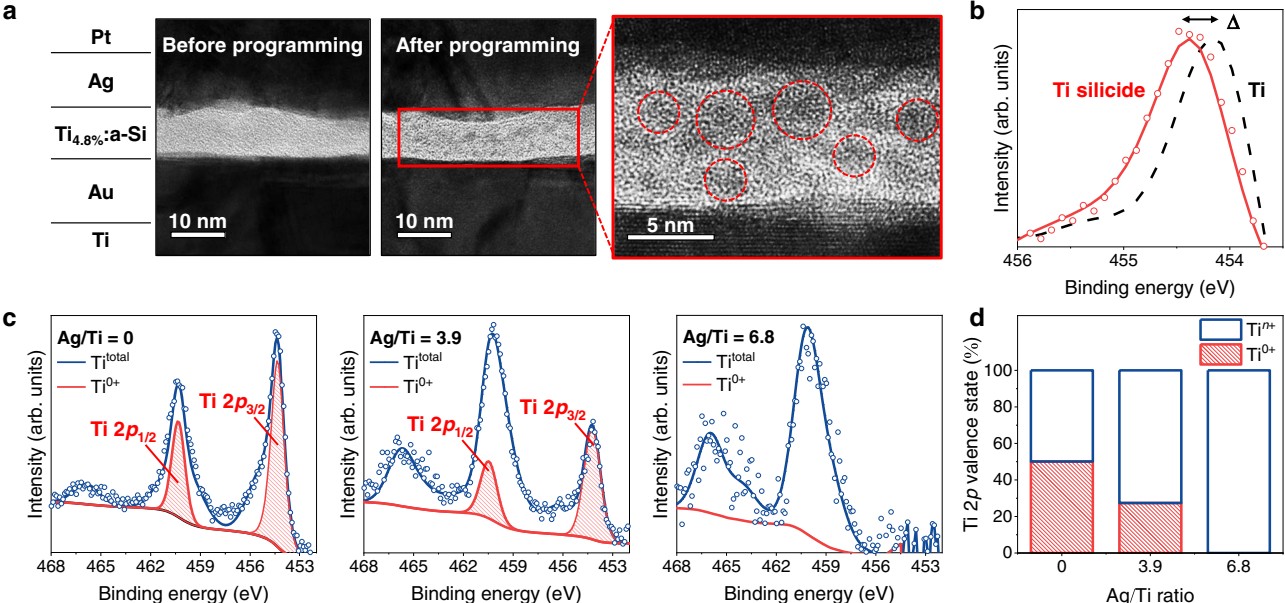

**Fig. 3 Microstructure and chemical composition analysis of the Ti$_{4.8\%}$:a-Si memristor. a** Cross-sectional transmission electron microscopy (TEM) image of the Ti$_{4.8\%}$:a-Si memristor taken before/after programming (right inset: high-resolution image taken after programming). Red circles indicate Ag-clusters nucleated inside the Ti$_{4.8\%}$:a-Si switching layer. **b** X-ray photoelectron spectroscopy (XPS) spectra of Ti $2p_{3/2}$ for Ti$_{4.8\%}$:a-Si (red) and metallic Ti (black) films indicating the presence of Ti silicide. **c** XPS spectra of Ti $2p$ for Ag-Ti-Si co-deposited films with Ag/Ti ratios of 0, 3.9, and 6.8. The blue and red lines indicate the total Ti spectrum and the de-convoluted Ti$^{0+}$ spectrum, respectively. As the Ag portion increases, Ti$^{0+}$ states change to Ti$^{n+}$. **d** The Ti$^{0+}$ and Ti$^{n+}$ (Ti$^{n+}$ = Ti$^{total}$ − Ti$^{0+}$) valence state distribution of the Ti $2p$ spectra for the Ag-Ti-Si films with 0, 3.9, and 6.8 Ag/Ti ratios.

resolved to Ag (200) crystal planes (Supplementary Fig. 11). These images demonstrate that the active electrode has been oxidised and inserted into a switching medium to increase the device conductance, facilitated by electrical pulses. Unlike conventional CBRAMs, the Ag-clusters are not accumulated on a counter electrode interface but rather nucleated within the Si matrix due to the electrochemical reduction of migrating Ag cations promoted by Ti nanoparticles. The same bright-field TEM imaging of the a-Si (pristine) device was also performed, and the visible filament protrusion was created at the inert electrode interface, shortening the effective distance between the two electrodes (Supplementary Fig. 10). We additionally certified the conditions for Ti to reliably operate as a reducing agent for Ag cations. First, the presence of Ti silicide in the Ti$_{4.8\%}$:a-Si device was confirmed by XPS measurements (Fig. 3b). We deposited a Ti$_{4.8\%}$:a-Si film that was deposited in the same condition as the device fabrication procedure and compared the Ti $2p_{3/2}$ spectrum with the metallic Ti film. The Ti $2p_{3/2}$ maximum of Ti$_{4.8\%}$:a-Si sample certainly shifted by 0.3–0.4 eV from metallic Ti $2p_{3/2}$ maximum, a general chemical shift observed between Ti silicide and metallic Ti[32,49]. The XPS characterisation technique has been used in subsequent studies for the electron transfer process from Ti to Ag. As shown in Fig. 3c, Ti$^{0+}$ clusters (red area) were found to be successfully oxidised to Ti$^{n+}$ states, as the Ag/Ti ratio increased from 0 to 6.8 in Ag-Ti-Si co-sputtered films that have been deposited in the same condition as the device fabrication procedure. Without the incorporation of Ag, almost 50.1% of the Ti silicide clusters existed in the Ti$^{0+}$ state; however, none of the Ti$^{0+}$ counts was discovered in the Ag/Ti = 6.8 ratio sample (Fig. 3d). These XPS results denote the capability of Ti as a reducing agent for Ag, owing to its low $E^{\varphi}$.

**Correlation between the reduction activity and the extent of analogue linearity improvement.** In Fig. 4, we tested different metals incorporated into the a-Si (densified) CBRAM to reinforce

our theory about analogue performance improvement by efficiently reducing Ag cations. We investigated Pt, W, and Ta binary compound metals of silicon with moderately different $E^{\varphi}$ values (Fig. 4a)[30] in order to identify the tendency between $E^{\varphi}$ and analogue linearity performance experimentally. After setting equivalent initial conductance, we characterised the analogue switching properties (1/−1 V, 1 μs). As shown in Fig. 4b, we referenced the previous data of a-Si (densified) and Ti$_{4.8\%}$:a-Si at 1/−1 V pulse condition and observed a direct contrast in switching linearity through M:a-Si (M = Pt, W, Ta, and Ti) devices. First, the Pt has a very high (positive) $E^{\varphi}$ value ($E^{\varphi}_{Pt}$ = 1.18 V) than Ag, prohibiting the electron transfer from Pt-clusters to Ag. This is completely opposite to our postulation, where the inserted metal clusters must act as a reducing agent for Ag cations. Thus, as shown in Fig. 4c, d, the average $v$ of the Pt:a-Si device was similar to or even worse than that of the a-Si (densified) device for both P/D. On the other hand, W and Ta show lower $E^{\varphi}$ values than Ag ($E^{\varphi}_{W}$ = 0.1 V and $E^{\varphi}_{Ta}$ = −0.6 V), e.g. $\Delta E^{\varphi}$ values of 0.7 V ($\Delta E^{\varphi}_{Ag-W}$) and 1.4 V ($\Delta E^{\varphi}_{Ag-Ta}$), respectively. Hence, such metals can serve as effective reducing agents, transferring electrons from metal clusters to Ag cations as in the Ti$_{4.8\%}$:a-Si device, where they demonstrated better linearity than a-Si (densified) device. Interestingly, from the results using various metal elements, the average $v$ from the ten-cycle showed a clear relationship with $\Delta E^{\varphi}$, where the larger $\Delta E^{\varphi}$ (Pt < W < Ta < Ti) produced much-improved linearity (Fig. 4c, d and Supplementary Fig. 14 for ten-P/D-cycle). The trend again confirms the validity of our idea, and more importantly, it implies that the linearity is now an adjustable parameter—from a more abrupt switching to a more gradual update depending on the $E^{\varphi}$ of inserted metals. In terms of analogue data retention of M:a-Si (M = Pt, W, and Ta) devices, Pt, W, and Ta metals exhibit a large degree of thermodynamic instability with Ag[50–52], which resulted in poor retention compared to Ti$_{4.8\%}$:a-Si device (Fig. 4e). The increased data stability through reduced interfacial energy between Si and Ag-Ti alloy could not be found in M:a-Si (M = Pt,

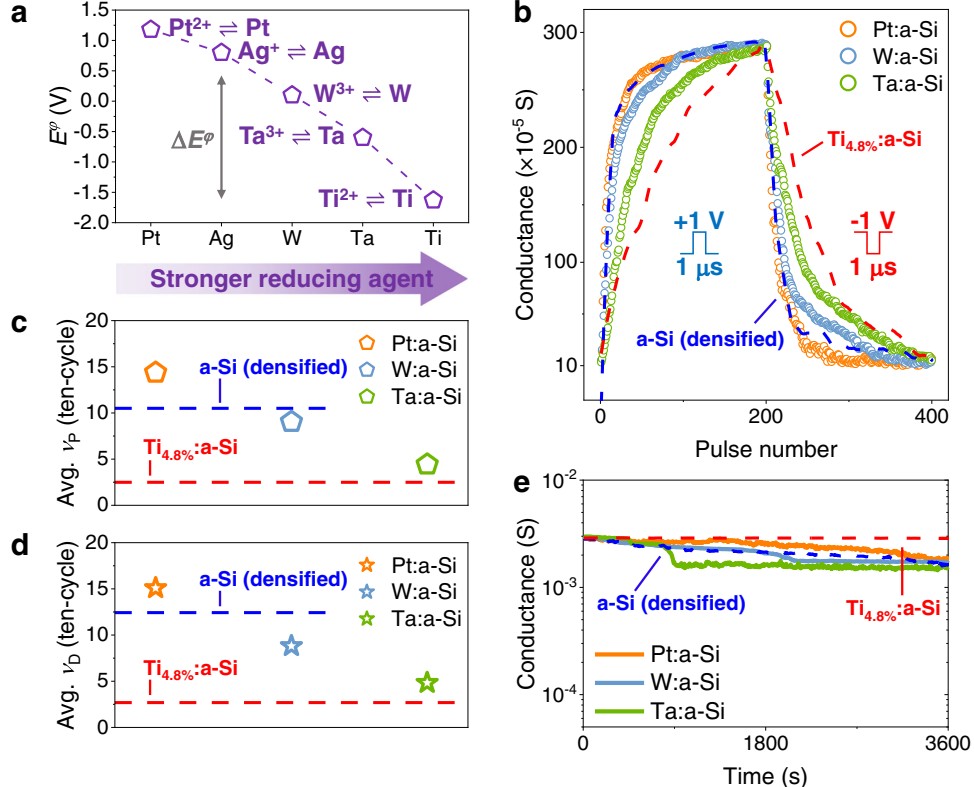

**Fig. 4 Analogue switching performance of the M:a-Si memristors in various metal systems (M = Pt, W, and Ta). a** Standard reduction potential of incorporated metals ($E^{\varphi}$ values taken from ref. [30]). **b** Analogue conductance update of the M:a-Si (M = Pt, W, and Ta) memristors under the $1/-1$ V, 1 μs pulse condition with 0.1 V, 1 μs read pulse. The P/D switching under the same pulse condition of the a-Si (densified) (dashed blue line) and $Ti_{4.8\%}$:a-Si (dashed red line) memristors are referenced. **c** Average potentiation and **d** depression nonlinearity factor of the M:a-Si (M = Pt, W, Ta, and Ti) memristors for ten-cycle under $1/-1$ V, 1 μs pulse condition. **e** Analogue data retention capability of the M:a-Si (M = Pt, W, and Ta) memristors, measured by 0.1 V read bias. The devices were programmed to the maximum conductance state ($G_{max}$) under $1/-1$ V, 1 μs pulse condition before the retention test. The retention test of the a-Si (densified) and $Ti_{4.8\%}$:a-Si memristors are referenced.

W, and Ta) devices. Furthermore, M:a-Si (M = Pt, W, and Ta) devices without an Ag electrode layer do not exhibit resistive switching in multiple *I-V* sweeps (Supplementary Fig. 15), which again guarantees that Ag is the only mobile element in the Si matrix when applying around $1/-1$ V. Following our design criteria, Ti was the ideal element to drive cluster-based analogue switching regarding low $E^{\varphi}$, fully miscible with Si and Ag.

**Feature extraction task using sparse coding with the experimentally measured analogue characteristics of memristor device.** Based on the experimental nonlinearity values extracted by the nonlinearity value calculation process (see Methods for details) of our memristor devices, we simulated a feature extraction task that consists of two stages: training features in receptive fields by stochastic gradient descent (SGD) algorithm and identifying a sparse representation of trained features using the locally competitive algorithm (LCA) (see Methods for simulation details)[53]. Sparse coding is a powerful algorithm reducing data dimension by extracting principal features. The algorithm underlies higher-level cognitive functions of biological neural systems[54] and is highly compatible with the crossbar array[11]. In Fig. 5a, the images used for training the receptive fields, the trained receptive fields with the ideal ($\nu = 0$) case, and the $Ti_{4.8\%}$:a-Si device case are presented. Throughout the receptive field learning by SGD, a learning rate parameter ($\beta$) value of 0.0004 is used. A feature extraction task is then performed using these receptive fields, and after separating the natural colour

image using a red-green-blue (RGB) filter, the LCA is applied (Fig. 5b). During this task, the $140 \times 140$ resolution natural image is broken into $20 \times 20$ pixel patches each. The time constant parameter ($\tau$) value of 0.008 is used during the feature extraction task by LCA. With these parameter settings, we compared the sparsity ($L_0$-norm, number of active neurons) and mean squared error (MSE) results for various threshold ($\lambda$) values. Here, the optimal result of the sparse coding application is conditioned by high sparsity (low $L_0$-norm) and low MSE. At the beginning of the iteration, membrane potentials of the neurons oscillated due to the competition of similar features; however, in the end, only a few features became winners and stabilised the final sparsity (Fig. 5c). MSE results are summarised in Fig. 5d, where the $Ti_{4.8\%}$:a-Si device case successfully demonstrated much higher reconstruction accuracy than that of the a-Si (pristine) device case, which is close to an ideal case. The reconstructed input images using the sparse codes and feature matrix are shown in Fig. 5e, where the $Ti_{4.8\%}$:a-Si device case can reproduce comparable $L_0$-norm and MSE images to the ideal case at the identical $\lambda$ value. We further verified the excellence of our $Ti_{4.8\%}$:a-Si device by a classification simulation using the Modified National Institute of Standards and Technology (MNIST) dataset (Supplementary Fig. 18).

## Discussion

In this study, we suggest a cluster-type CBRAM with highly enhanced synapse characteristics. The maximised reduction

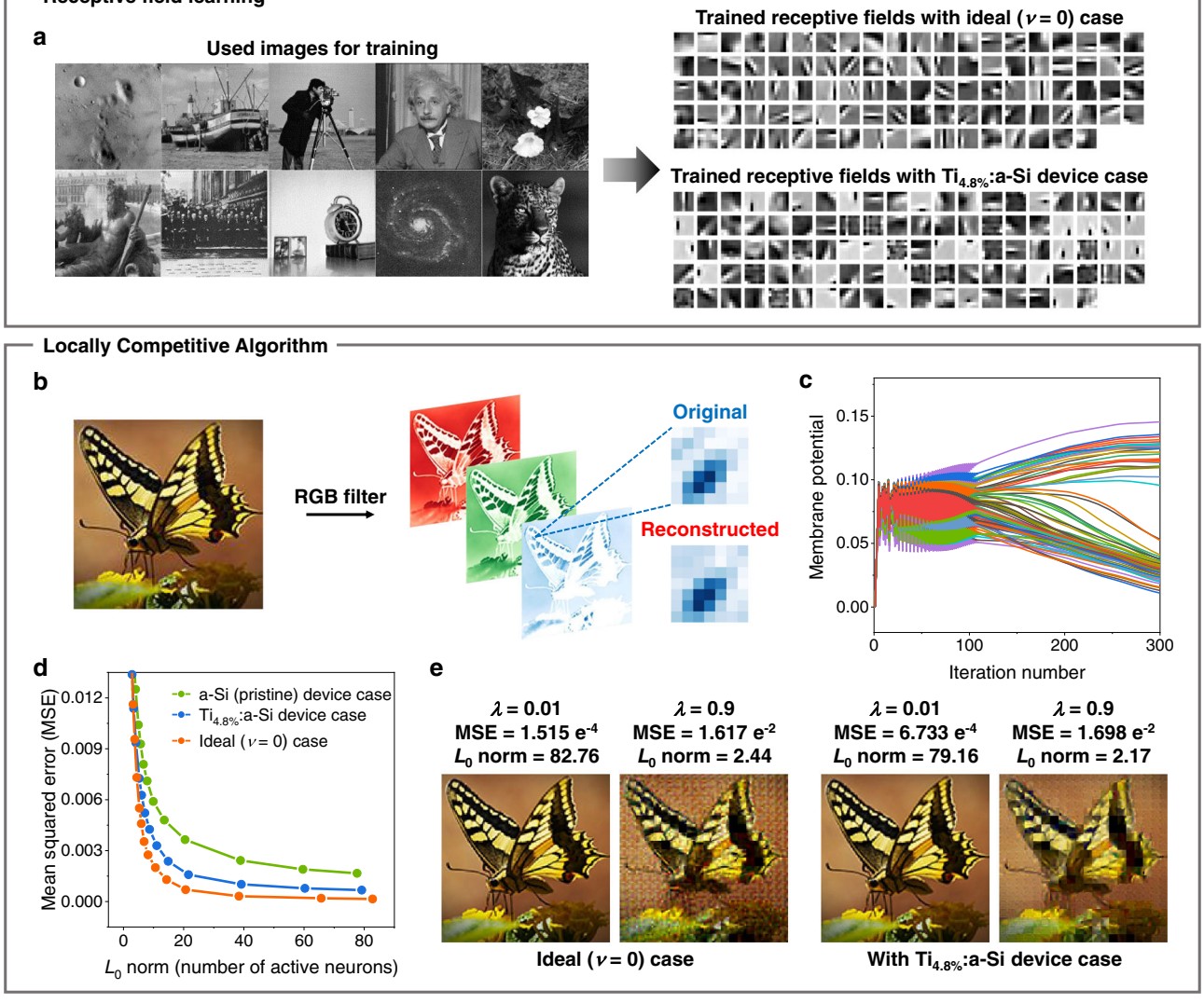

**Fig. 5 Feature extraction task of sparse coding simulation using the measured device characteristics. a** Receptive field trained through the stochastic gradient descent (SGD) and locally competitive algorithm (LCA) after 50 iterations. Ten natural images are used for training receptive fields (left). Trained features, similar to Garbor filters, are shown for the ideal ($\nu = 0$) case and the Ti$_{4.8\%}$:a-Si device case. **b** The original natural colour image (140 × 140 resolution image) passed to a red-green-blue (RGB) filter to be encoded and reconstructed through the LCA. After reconstruction, each RGB patch image is combined to a natural colour image. **c** Membrane potential of output neurons during iterations. **d** Reconstruction mean squared error (MSE) of each memristor device case in different sparsity ($L_0$ norm). **e** Reconstructed images after sparse coding simulation for the ideal case and the Ti$_{4.8\%}$:a-Si device case with threshold, $\lambda = 0.01/0.9$, and resultant $L_0$ norm.

probability of Ag cations inside the a-Si enables the formation of clusters instead of conducting filament and helps relieve the linearity and on/off ratio trade-off issue, allowing linear potentiation/depression and a large conductance range (~244). In addition, the intermetallic compound formation with the Ag filament further accomplishes long-term data stability (~99% at 1 h). The degree of electrochemical interactivity between Ag cation and silicide metals was well controlled through the manageable physical parameter—$E^\varphi$ values of inserted metals (Pt, W, Ta, and Ti)—which paves the way to tailoring the desirable linearity for various applications. With these analogue performances, the Ti$_{4.8\%}$:a-Si device functioned well in memristor-based image processing algorithms. We conclude by noting that the extent of reduction activity is closely related to the analogue linearity performance. We believe our results further broaden our fundamental understanding of the resistive switching mechanism in CBRAM. In addition, our engineering strategy is capable of being applied to other

memristors with different material systems to achieve high-performance in neuromorphic computing applications.

## Methods

**Device fabrication**. The two-terminal metal-insulator-metal crossbar memristor devices were fabricated on a p-type (100) Si wafer with 200 nm SiO$_2$. All layers were crossbar patterned with the double-layered photoresist (LOR 2 A + AZ 5214 E) by photolithography with $4 \times 4$ μm$^2$ in cell size. The bottom electrode, composed of 3 nm Ti adhesion layer and 27 nm Au layer, was deposited by electron-beam evaporation on Si/SiO$_2$ substrate, followed by a lift-off process with acetone and developer (AZ 300 MIF). After switching layer patterning, the 7 nm thick pristine a-Si film was deposited by radio frequency (RF) sputtering a Si target at 20 W and room substrate temperature. The densified a-Si film was deposited by RF sputtering an identical Si target at 70 W and 350 °C substrate temperature with a post rapid thermal annealing (RTA) process at 350 °C for 5 min in the Ar atmosphere. The M:a-Si (M = Pt, W, Ta, and Ti) layers were deposited by RF co-sputtering a-Si with M targets at 70 W of Si and 15 W of M, keeping the 350 °C deposition temperature. The identical post RTA process was also performed in M:a-Si layers. In all devices, the switching layers of the memristors were observed

to keep the 7 nm thickness, and the post RTA process was performed after the lift-off process with Microresist, mr-REM 700 solution. Without breaking the vacuum, the top electrode of 10 nm of Ag and 40 nm of the protective layer of Pt were deposited with RF sputtering the Ag and Pt targets at 30 W, with a subsequent lift-off process in mr-REM 700. All the RF sputtering processes were done after reaching a base pressure of $2 \times 10^{-7}$ Torr or less and under a working pressure of 3 mTorr in the Ar atmosphere.

**Electrical measurement**. Quasi-static current-voltage (*I-V*) measurements were performed by Keithley 4200A-SCS with a source measure unit (SMU). The repetitive voltage bias was applied to the crossbar patterned top electrode while the bottom electrode was grounded. The compliance current was set to 1 mA during the forming and set *I-V* sweeps. Analogue retention characteristics were carried out by a read voltage of 0.1 V after applying potentiation pulses. Analogue switching characteristics of memristors were executed in Keithley 4200A-SCS with a pulse measure unit (PMU). The devices were pre-formed before the analogue measurement. All programming and read pulse widths were 1 μs, and potentiation and depression pulse amplitudes were set identically with opposite polarity, ranging from 0.6/−0.6 to 1.1/−1.1 V. The read voltage pulse of 0.1 V was directly applied after programming pulses to measure the conductance update without affecting the conductance state of the memristor.

**Nonlinearity value calculation**. The ideal linearity is defined as a state where the change in conductance update due to potentiation/depression pulse does not depend on the current conductance state of the device. To quantify such linear characteristics in potentiation and depression, the nonlinearity values (*v*) were extracted using the following equations[55]:

$$G_P = G_1 \left(1 - e^{-\nu_P N}\right) + G_{min} \tag{1}$$

$$G_D = G_{max} - G_1 \left[1 - e^{-\nu_D \left(N_{max} - N\right)}\right] \tag{2}$$

$$G_1 = \frac{G_{max} - G_{min}}{1 - e^{-\nu N_{max}}} \tag{3}$$

Here, $G_P$ and $G_D$ are the conductance for each potentiation and depression given by the above equations. $G_{max}$ and $G_{min}$ are the maximum and minimum conductance states, respectively. $N$ and $N_{max}$ are the normalised pulse number and maximum normalised pulse number, respectively, the latter of which is 1. $G_1$ is the function of *v* in order to fit (normalise) the $G_P$ and $G_D$ functions within the range of $G_{max}$, $G_{min}$, and $N_{max}$. The nonlinearity factors for potentiation ($\nu_P$) and depression ($\nu_D$) are calculated by fitting the above equations through minimising the absolute difference between the fitting curve and the experimental results at every pulse. At $v = 0$, the conductance update is ideally linear. As the *v* increases, the conductance rapidly saturates to $G_{max}$ even at a small number of potentiation pulses, and the opposite situation occurs for depression pulses.

**Device characterisation**. To adjust the film thickness, Alpha-step IQ surface profiler was used to measure the step height during the deposition, and the final device thickness was determined by transmission electron microscopy (TEM) measurements. The density of a-Si (pristine) and a-Si (densified) films were characterised by X-ray reflectometry measurement (ATX-G, Rigaku, operated at 40 kV, 250 mA), and the spectra were collected using Cu $K_\alpha$ x-ray source ($\lambda = 1.54$ Å) with a scan range of 0–6 degrees in 2θ. X-ray photoelectron spectra (Nexsa, ThermoFisher Scientific) on Ti, Ti-Si, and Ag-Ti-Si films were measured using a micro-focus monochromatic Al $K_\alpha$ X-ray source ($hv = 1486.6$ eV). The Ti-Si and Ag-Ti-Si films were deposited in the same procedure with the device fabrication, including the post RTA process at 350 °C for 5 min in the Ar atmosphere. The amorphous phase of a-Si (densified) and $Ti_{4.8\%}$:a-Si films were characterised by X-ray diffraction measurement (Dmax2500-PC, Rigaku, operated at 40 kV, 200 mA) using glancing incident scan mode with a scan range of 1 degree and a scan speed of 2 degree/min.

**Transmission electron microscopy measurement**. Electron microscopy specimens were prepared by a focused ion beam (FIB) (Helios NanoLab 600) system along with the scanning electron microscopy imaging. For programmed cross-sectional images, the memristor device was programmed before FIB sampling with potentiation programming pulses. Pt was deposited to protect the specimen surface, using a "C" gas injection system for electron beam deposition and ion beam deposition. The samples were rough milled and lifted out using a probe system to attach to a TEM grid, and the specimens were thinned and fine milled to 100 nm. The micro-images of the prepared samples were investigated by a field-emission TEM (Technai F20 G2, FEI), and scanning-TEM (STEM) images were obtained using a $C_s$-corrected STEM (TitanTM 80-300, FEI) equipped with a fast charged-coupled device camera (Gatan, Oneview 1095).

**Feature extraction task**. The sparse coding simulation is comprised of two stages: receptive field learning and the following feature extraction step to sparsely

represent an input image. First, in the feature training step, we used stochastic gradient descent (SGD), one of the back-propagation algorithms, and a locally competitive algorithm (LCA) to train the dictionaries that create Gabor-like receptive fields. In detail, we selected primitive features for each input using LCA, and the activated receptive fields were only adapted while computing the error gradient by SGD. The error gradient learning rule was implemented using the following equations[56]:

$$\nabla E = -\left(X - a \cdot \Phi^T\right) \otimes a \tag{4}$$

$$\triangle \Phi^T = \beta \left(X - a \cdot \Phi^T\right) \otimes a \tag{5}$$

Here, *E* is the error with respect to the receptive fields, Φ is the matrix of receptive fields from the conductance values of the memristor, *a* is the activities of the neurons by LCA, *X* is the original input vector, and *β* is the learning rate ($\beta = 0.0004$ is used). We applied the nonlinearity values of each memristor device, a-Si (pristine) and $Ti_{4.8\%}$:a-Si, measured at 0.6/−0.6 V P/D pulse condition and compared with the ideal device case ($v = 0$).

Then, feature extraction by sparse coding was performed using LCA after the training. LCA was thus applied twice in our simulation. LCA is mathematically expressed with the following equations[53]:

$$\frac{du}{dt} = \frac{1}{\tau} \left[-u + \left(X - a \cdot \Phi^T\right) \cdot \Phi + a\right] \tag{6}$$

$$a = T(u, \lambda) = \begin{cases} u, & if\ |u| \geq \lambda \\ 4u - 3\lambda, & if\ 0.75\lambda < |u| < \lambda \\ 0, & if\ |u| \leq 0.75\lambda \end{cases} \tag{7}$$

Here, *u* is the membrane potential of the output neurons, and *a* is expressed by a threshold function $T(u, \lambda)$. During the algorithm process, the membrane potential of output neurons is governed by the $X \cdot \Phi$ term, which indicates the closeness between the input and output neuron elements. The important feature of LCA is a $-a \cdot \Phi^T \Phi$ term that prevents similar receptive fields from being simultaneously activated and ensures proper sparseness of the network. Through this inhibition term in LCA, the network interactively triggers the competition of active neurons and finds the optimal sparsity, which is considered an essential aspect of the biological nervous system[54]. Further, the leakage term $-u$ continuously impacts the membrane potential of the output neurons with the time constant *τ* ($\tau = 0.008$ is used). After the network stabilisation, the optimised sparsity of active neurons is obtained, and with the linear combination of optimised receptive fields, the sparse code can be used to reconstruct the original input patterns. Finally, the reconstruction error of the algorithm is calculated by comparing the original image with the reconstructed one. The mean squared error (MSE) is calculated with the following equation[56]:

$$MSE = \frac{1}{400} \sum_{i=1}^{400} (X - a \cdot \Phi^T)_i \tag{8}$$

## Data availability

The data that support the findings of this study are present in the paper and/or the Supplementary Information. The experimental raw data, e.g. electrical measurement, device characterisation, and sparse coding simulation, are too large to be shared publicly. The corresponding author will directly respond to any additional data request.

## Code availability

The code that supports the results of the sparse coding application is available in the GitHub repository https://github.com/kykykkkk/LCA.

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

## Acknowledgements

This work was supported in part by the Korea Institute of Science and Technology (Grant no. 2E31550) and National R&D Program through the National Research Foundation of Korea (NRF) funded by Ministry of Science and ICT (Grant nos. 2021M3F3A2A01037814, 2021M3F3A2A01037738, and 2021M3F3A2A03017782).

## Author contributions

Y.J.J. directed and supported this project. J.H.K. designed the experiments, including the sample fabrication with S.M.H., J.K.P., and I.H.K., measurements with S.Y.L., and analyses with S.B.K., J.Y.K., and J.K.P.. T.Y.K. and J.W.K. conducted the simulations. J.H.K. and Y.J.J. wrote the manuscript and all authors were involved in the discussion of the results and commented on the manuscript.

## Competing interests

The authors declare no competing interests.
