## [Peer Review File · Nature Communications]

Title: Cluster-type analogue memristor by engineering redox dynamics for high-performance neuromorphic computingREVIEWER COMMENTS

Reviewer #1 (Remarks to the Author):

Review on the manuscript (NCOMMS-21-46755-T) entitled “Cluster-type analogue memristor by engineering redox dynamics for high performance neuromorphic computing”

In this manuscript, authors propose a cluster-type analogue CBRAM in which the switching is dependent on the amount of Ag-clusters rather than filament formation. Authors have demonstrated their work in a magnificent way. However, reviewer feels that CBRAM has been proposed long back. Only the new concept that authors highlight is engineering the redox dynamics. It is very much important to justify how their work is different from already existing literature. Reviewer reject this manuscript for the publication.

Reviewer #2 (Remarks to the Author):

In this work, the authors demonstrated Ti-based cluster-type 18 analogue memristors. For the following reasons, Ti is entrenched in amorphous densified silicon: low reduction potential, thermodynamic miscibility with silicon, and alloy formation with amorphous alumina. As a result of the electrochemical reduction activity of Ag cations induced by these Ti clusters, an extraordinarily linear potentiation/depression may be achieved with a wide conductance range (244) and extended data retention (99 percent at 1 hour). In addition, it is interesting to see the effect of reducing agents on analog switching behaviors. I believe there are sufficient information and enough evidence of how analog switching behaviors have been improved by subtly tuning additional metal atoms to silver active metal. I would like to recommend the publication of this work when the authors can provide below details.

“The authors show successful demonstration of small scale memristors with a 4 micrometers junction. Since CBRAM involves the high thermal process, the size of junctions may affect the device’s switching behaviors, which can be a huge limiting factor for scalability. I would like to know whether the authors explored the effect of junction size.”

“To fully implement analog switching device for the next generation computing machine, the retention at a low-mid conductance level is significant rather than one at high conductance level. Although authors’ analog memristor show great performance in terms of linearity, on-off ratio, and high conductance level retention, retention at low- and mid-level conductance level is not presented. As the low- and mid-level retention can be more fluctuating or decaying due to the weak filament formation, the authors should be able to give specific numbers for the future study.”

Reviewer #3 (Remarks to the Author):

In this paper the authors present, based on fundamental considerations in materials research, the development of cluster-type memristors, which are characterized in particular by a high degree of linearity in the switching behavior and an increased conductance range compared to such elements whose conduction mechanisms are based on the dynamics of filaments. The content is timely and clearly written but needs a revision according to the following points. Were other pulse amplitudes used, with what results. The authors refer to DC measurements; were these actually DC experiments or quasi-static measurements. Were the realized memristors characterized with respect to the elementary properties mentioned by Leon Chua? Is further linearization possible, physical limits? Please mention also device variations for these memristors.

Page 2: The memristor was not rediscovered in 2008 but experimental results of a fabricated device have been related to the continuously developed theory of Leon Chua starting in 1971.

Page 6: The authors refer to an optimization that is usually done mathematically but here they are referring to a material selection with different values of E^ϕ values. This point may be better clarified.

Page 10: The authors should explain and specify the “experimental model”

Page 10 The sparse coding applications – including settings and parameters - should be described more detailed.

Fig. 1a: The blue graphics are confusing.

Fig. 1b-d: Please describe the dynamics in the figure caption.

Fig. 2b: The underlying curve might look rather nonlinear?

Page 18: G_1 is not defined

Define all quantities used in the manuscript, also when known in the community of the authors, e.g. E^ϕ . Finally, the paper should be checked by a native speaker for improvement.

AUTHORS' RESPONSE TO THE REFEREES' COMMENTS

We greatly appreciate the editor and all the referees for generously lending us help and sharing their valuable insights on our research. After careful consideration, we have carried out further experiments and corrected the manuscript according to the referees' suggestions. In the following letter, we provide point-by-point responses to the referees' comments. The changes are indicated as an **orange color** in both revised manuscript and response letter. We sincerely hope that our responses are satisfactory to the editor and to all the referees.

Reviewer #1

General comment

In this manuscript, authors propose a cluster-type analogue CBRAM in which the switching is dependent on the amount of Ag-clusters rather than filament formation. Authors have demonstrated their work in a magnificent way. However, reviewer feels that CBRAM has been proposed long back. Only the new concept that authors highlight is engineering the redox dynamics. It is very much important to justify how their work is different from already existing literature. Reviewer reject this manuscript for the publication.

Response

We sincerely appreciate the reviewer for the critical evaluation, which made us ruminant on the motivation of our research. Indeed, we agree with the reviewer that the concept of memristor was first designed by Chua in 1971¹, and the resistive switching effect of the insulating layer was discovered in the 1960s². The Response Fig. 1 displays the number of publication history each year from 1973 to 2021 for keywords, ‘Resistive switching,’ ‘RRAM,’ ‘Memristive device,’ and Neuromorphic RRAM,’ respectively. Therefore, this kind of resistive switching device has been widely studied and used as a memory application known as resistive random-access memory (RRAM) for a long time. However, it is in very recent times that these RRAM devices have been applied in neuromorphic computing applications as an artificial synapse. As shown in Response Fig. 1, the publications regarding ‘memristive device or neuromorphic RRAM’ have mostly been published after 2010 and numerous developments including resistive switching mechanism³⁻⁷, improved switching performance⁸⁻¹¹, and diverse computing or biological applications¹²⁻¹⁵ have been made to fulfill the key requirements as a synapse device for neuromorphic computing. Unlike the conventional RRAM memory application based on digital-like resistive switching, the neuromorphic computing requires the analogue device features such as linear conductance update, large on/off ratio, and reliable retention to increase the accuracy, convergence effectiveness, and data maintenance of neural networks, respectively¹⁶. However, even in neuromorphic application studies conducted since 2010, there has been a lack of understanding regarding the analogue characteristic mechanism of memristive devices needed to enhance the analogue performance.

Response Fig. 1 The number of publications each year (1973 to 2021) for keywords, ‘Resistive switching,’ ‘RRAM,’ ‘Memristive device,’ and Neuromorphic RRAM,’ respectively. The data source is from Dimensions.ai, a research information system that covers publications, books, and conferences provided by Digital Science (<https://www.dimensions.ai/>).

Among memristive devices, CBRAM has exhibited outstanding resistive switching ability under electric field bias and is named after the basic operation mechanism where metallic filament bridges the anode and cathode. To date, CBRAM’s fundamental filament formation mechanism has been studied regarding the electrochemical metal redox reactions. Although many parameters of the filament dynamics have been identified, such as electrode reactions based on charge transfer overpotential⁶, there has been a lack of research on the causality of analogue characteristic changes caused by the modulations among those parameters. The details were missing, specifically regarding engineering adequate filament condition, which would be practical for the analogue synapse applications. Consequently, most CBRAMs are used with the same filamentary switching mechanism that have been used for conventional memory device application and the device showed digital-like potentiation/depression switching or only managed on/off ratios typically less than 10 in the analogue switching region¹⁷. A few research achieved a large analogue dynamic range by

applying large voltage pulses or pulse numbers, but in such case, the analogue linearity critically deteriorated^{10,18,19}. This trade-off relationship strongly originates from the positive feedback effect in an electric field during the filament growth process, where the electric field induces exponential ionic migration toward the depleted filament region^{6,20,21}. Numerous attempts have been made to improve CBRAM's analogue performance; switching variation has been lessened by the one-dimensional filament confinement effect¹⁰, sub-femtojoule power consumption has been made by atomically thin filament¹¹, and retention time has been improved by the filament interfacial energy stabilization⁸, yet the trade-off between the two major analogue characteristics, the linearity and on/off ratio, has not been fully resolved. Therefore, the importance of our research can be summarized as in the following respect:

- 1) In this research, we presented a novel methodology that creates a cluster-type memristive device by engineering reduction probability of Ag cations; we added metal reducing agents and regulated the ion migration process, which minimized the trade-off issue between linearity and on/off ratio commonly observed in conventional filament-type CBRAMs (Response Fig. 2a). We observed the Ag clusters inside the Ti_{4.8%}:a-Si layer from TEM images (Response Fig. 2b). As far as we know, our work distinguishes from other research in that certain clues or physical parameter have rarely been suggested in terms of controlling the analogue linearity of synaptic devices.

Response Fig. 2 a Average nonlinearity factor and on/off ratio of ten-P/D-cycle under three different pulse amplitudes for a-Si (pristine), a-Si (densified), and Ti_{4.8%}:a-Si devices. **b** The

TEM images of the $\text{Ti}_{4.8\%}:\text{a-Si}$ device taken before/after programming, with the red circles indicating Ag-clusters nucleated inside the $\text{Ti}_{4.8\%}:\text{a-Si}$ switching layer and corresponding fast Fourier transform results.

- 2) To enable high ion reduction probability, two types of mechanisms are utilized: first, the longer temporal margin for capturing free electrons injected from the cathode, and second, thermodynamically preferred electron transfer from Ti to Ag cations due to the large reduction potential difference between Ti and Ag. As shown in Response Fig. 3a, the $\text{Ti}_{4.8\%}:\text{a-Si}$ device demonstrated a highly linear potentiation/depression along with a large conductance range of 244. To our best knowledge, the two key analogue features, the linear switching and the dynamic range of our optimized device, exhibit a significant deviation from the previously reported analogue memristors (Response Fig. 3b). Moreover, owing to the Ag-Ti alloy formation property, the data retention time has been improved through the filament interfacial energy stabilization (Response Fig. 3c).

Response Fig. 3 a Linear conductance update in the full analogue range under the 1.1/-1.1 V, 1 μs pulse condition with 0.1 V, 1 μs read pulse. **b** Benchmark of the $\text{Ti}_{4.8\%}:\text{a-Si}$ memristor with previously reported Ag- and Cu-based conductive-bridge random access memory (CBRAM) devices for linearity and on/off ratio characteristics (ref. ^{9,10,15,17,18,33-45} in the revised manuscript). **c** Improved analogue data retention capability of the $\text{Ti}_{4.8\%}:\text{a-Si}$ device compared to the a-Si (pristine) and a-Si (densified) devices, measured by 0.1 V read bias.

- 3) The impact of our work also lies in our demonstration that the device parameter, analogue linearity, is selectively adjustable through the manageable physical parameter, reduction potential of transition-metal. The proposed mechanism applied well even when using other metals with different reduction potentials and different

alloy abilities with Ag, such as Ta, W, and Pt (Response Fig. 4a,c). Above all, as shown in Response Fig. 4b, the correlation between the extent of reduction activity and analogue linearity is now proven.

Response Fig. 4 **a** Standard reduction potential of incorporated metals (Pt, W, Ta, and Ti) **b** Analogue conductance update of the M:a-Si (M = Pt, W, and Ta) memristors under the 1/-1 V, 1 μ s pulse condition with 0.1 V, 1 μ s read pulse. The P/D switching under the same pulse condition of the a-Si (densified) (dashed blue line) and $Ti_{4.8\%}$:a-Si (dashed red line) devices are referenced. **c** Analogue data retention capability of the M:a-Si (M = Pt, W, and Ta) devices, measured by 0.1 V read bias. The retention test of the a-Si (densified) and $Ti_{4.8\%}$:a-Si devices is referenced.

Lastly, we conclude that our study distinguishes from previous research concerning the three aspects described above. Our proposed CBRAM device design criteria considering reduction potential, thermodynamic miscibility with switching matrix, and alloy formation with mobile cation, may pave the way to the applications in other CBRAMs or memristive systems and thus, achieve high-performance neuromorphic computing. Therefore, we deeply express our gratitude to the reviewer for participating as the referee to our work and would highly appreciate it should the reviewer reconsider our paper.

Reviewer #2

General comment

In this work, the authors demonstrated Ti-based cluster-type 18 analogue memristors. For the following reasons, Ti is entrenched in amorphous densified silicon: low reduction potential, thermodynamic miscibility with silicon, and alloy formation with amorphous alumina. As a result of the electrochemical reduction activity of Ag cations induced by these Ti clusters, an extraordinarily linear potentiation/depression may be achieved with a wide conductance range (244) and extended data retention (99 percent at 1 hour). In addition, it is interesting to see the effect of reducing agents on analog switching behaviors. I believe there are sufficient information and enough evidence of how analog switching behaviors have been improved by subtly tuning additional metal atoms to silver active metal. I would like to recommend the publication of this work when the authors can provide below details.

Response

We immensely thank you for your favorable comments, highlighting concepts of our study as well as the results, e.g., “interesting to see the effect of reducing agents,” and having “sufficient information and enough evidence.” All your pertinent suggestions are critical and essential considerations in our research field, which we unfortunately did not present in our first manuscript submission. Following your valuable comments, we have made careful revisions and carried out further experiments. We hope that our revised manuscript and responses are satisfactory to you. Thank you again for reviewing our paper so as to make our research more constructive.

Comment #1

The authors show successful demonstration of small scale memristors with a 4 micrometers junction. Since CBRAM involves the high thermal process, the size of junctions may affect the device’s switching behaviors, which can be a huge limiting factor for scalability. I would like to know whether the authors explored the effect of junction size.

Response

Thank you for your keen insight about the junction size effect of the memristor device that may affect the device’s switching characteristics. In fact, as the reviewer mentioned, there

has been much research, including the 3D calculation of the local temperature of filaments²², that verifies the thermal effect, mostly Joule heating, affecting the switching process of memristive devices². Since it is vital to explore the scaling effect of junction size, we newly designed and fabricated the Ti_{4.8%}:a-Si device in nano- to micro-cell scale (100 nm ~ 2 μm) to be in a single chip.

First, the resistive switching characteristic of the Ti_{4.8%}:a-Si device was successfully measured after the forming process in all junction sizes of 2 μm, 600 nm, and 100 nm, as shown in Response Fig. 5a. As the junction size scaled down, the initial conductance (G_{initial}) level (before forming) decreased as $G_{\text{initial}_{2\ \mu\text{m}}} = 1.68 \times 10^{-6}\ \text{S}$, $G_{\text{initial}_{600\ \text{nm}}} = 1.06 \times 10^{-8}\ \text{S}$, and $G_{\text{initial}_{100\ \text{nm}}} = 1.94 \times 10^{-9}\ \text{S}$, which is since the initial current is proportional to the cell area, so as the G_{initial} level decreased as the junction size scaled down. However, the conductance range of the set/reset processes, along with the set/reset voltage, remained constant as the device junction size scaled down and therefore, we conclude that the effect of the device junction size in I - V switching curve was minuscule. Second, we applied 0.6/-0.6 V, 1 μs identical pulse train to assess whether the analogue performance deteriorates due to the thermal effect in nano-scale junction devices. The initial conductance levels (after forming) of the three different junction devices were fixed to 0.1 mS before the analogue measurement so as to ensure a fair comparison. In 2 μm junction cell, a linear conductance update ($v_P = 2.41$ and $v_D = 2.18$) and G_{max} near 1.5 mS, were achieved. Furthermore, the two nano-scale junction cells (600 and 100 nm) demonstrated similar nonlinearity values as micrometer junction devices, reaching G_{max} near 1.5 mS. Therefore, we also did not observe the degradation of analogue switching characteristics as the device junction size scaled down.

Response Fig. 5 The scaling effect of junction size in Ti_{4.8%}:a-Si device switching behavior (100 nm ~ 2 μm) **a** Quasi-static current-voltage (*I-V*) switching characteristics of the forming process with the following set and reset processes for the 2 μm, 600 nm, and 100 nm junction devices. **b** Analogue conductance update of 1 μs duration for the 2 μm, 600 nm, and 100 nm junction devices under 0.6/-0.6 V pulse condition with 1 μs duration for potentiation and depression.

As the reviewer mentioned, we deeply agree that the junction size effect is critical, but unfortunately, below 100 nm junction size was beyond our fabrication ability. The thermal effect in RRAM is known to happen very locally that Joule heating generates localized heat and affects the diffusion process of ions⁷. Besides, the phase change memory (PCM) that carries high temperature Joule heating-based switching process has already been realized in sub-nano-scale size by effectively preventing thermal dissipation through adding a capping layer between the top electrode and the switching material.²³ Therefore, we consider that the CBRAM may also reach nano-scale junction size by effectively utilizing the thermal heat since the electric field drift effect is relatively more dominant in the switching process than thermal diffusion, which is opposite to the PCM device². We believe that further CBRAM research regarding the junction size effect is pivotal to accelerating the industrialization of CBRAM devices. We are immensely thankful to the reviewer that led us to conduct device scaling experiments and demonstrate nano-scale junction size analogue Ti_{4.8%}:a-Si CBRAM.

Changes in Manuscript

- In the Supplementary Information of revised manuscript, Response Fig. 5 is included as Supplementary Fig. 9 along with figure caption and additional information.

Comment #2

To fully implement analog switching device for the next generation computing machine, the retention at a low-mid conductance level is significant rather than one at high conductance level. Although authors' analog memristor show great performance in terms of linearity, on-off ratio, and high conductance level retention, retention at low- and mid-level conductance level is not presented. As the low- and mid-level retention can be more fluctuating or decaying due to the weak filament formation, the authors should be able to give specific numbers for the future study.

Response

We express our sincere appreciation for pointing out the missing measurement, indispensable for the memristor to be fully implemented as neuromorphic synapse device. Indeed, to perform high-level inference tasks with memristor device, having reliable retention at multi-level conductance states is an essential characteristic⁸ as one of our key material selection criteria in this research was thermodynamic alloy formation to increase retention time⁸. Owing to Ag-Ti silicide alloy formation, the high interfacial energy of Ag in the Si matrix was stabilized and our Ti_{4.8%}:a-Si device outstood amongst other a-Si devices regarding the retention performance of 1.1% conductance decay after an hour. However, in the main manuscript, we only referred to maximum conductance level retention results even though low- and mid-level conductance level retention are extremely crucial, as you suggested. We appreciate the reviewer's great insight which guided us to conduct further experiments, and following your comment, we present the retention results at multi-level conductance state of Ti_{4.8%}:a-Si device with raised temperatures (Response Fig. 6b). Before the retention test, the device was programmed to a specific conductance state, and the temperature was raised. As you can see in the Response Fig. 6a, our Ti incorporated memristor sustained its initial conductance level for an hour, and the percentage of conductance decay remained less than 2% at all conductance levels. It is notable that the conductance decay results of low to high conductance levels show little difference, although the filament or clusters easily dissolve at low conductance level as the reviewer has mentioned. Furthermore, the device also

demonstrated stable data retention characteristics in all low-, mid-, and high-conductance levels with elevated temperatures at 100 °C, 150 °C, and 200 °C. However, as the temperature was raised to 200 °C, the conductance decay percentage slightly increased, which may be due to the accelerated Ag cluster dissolution process. We have added our interesting retention results to the main text and supplementary information, thanks to the reviewer’s keen suggestion that has greatly improved the quality of our research.

Response Fig. 6 Analogue data retention test with raised temperature of Ti_{4.8%}:a-Si device for 1 h. **a** The device is tested at various multi-conductance levels at room temperature and the conductance decay (%) at each conductance level. **b** The retention test of low-, mid-, and high-conductance levels with raised temperature at 100 °C, 150 °C, and 200 °C and the corresponding conductance decay (%) values. The devices were programmed to a specific conductance state before the retention test, measured by 0.1 V read bias.

Changes in Manuscript

- In the Supplementary Information of revised manuscript, Response Fig. 6 is included as Supplementary Fig. 9 along with figure caption and additional information.
- In the revised manuscript page 8, the relevant description of this results is added as **“More importantly, the Ti_{4.8%}:a-Si device sustained considerably low conductance decay percentage at multi-conductance levels at room temperature and**

demonstrated stable retention performance with elevated temperatures (100 °C, 150 °C, and 200 °C) at low-,mid-, and high-conductance levels (Supplementary Fig. 9).”

Reviewer #3

General comment

In this paper the authors present, based on fundamental considerations in materials research, the development of cluster-type memristors, which are characterized in particular by a high degree of linearity in the switching behavior and an increased conductance range compared to such elements whose conduction mechanisms are based on the dynamics of filaments. The content is timely and clearly written but needs a revision according to the following points.

Response

We sincerely appreciate your constructive comments that considerably helped to improve the quality of our work. Your valuable comments made us to address critical facts that we had not dealt with before and indeed inspired us to study and clearly mention the terminologies and experimental conditions used in our research. Based on your suggestions, we have made careful revisions and carried out further experiments. We wish that our revised manuscript and responses would be satisfactory. Thank you again for contributing to the improvement of our research.

Comment #1

Were other pulse amplitudes used, with what results.

Response

Thank you for your insightful comment. As the reviewer has pointed out, the pulse amplitude during potentiation/depression (P/D) measurement is indeed a critical factor in assessing the analogue characteristics of CBRAM. For most CBRAMs, the trade-off relationship between dynamic range (on/off ratio) and linearity performance (nonlinearity factor) is easily observed¹⁰, which mostly originates from the positive feedback in an electric field. In large pulse amplitude, it is more likely to be easily saturated to maximum conductance state at a small number of pulses with more digital-like resistance switching despite of the large dynamic range; in small pulse amplitude, the opposite situation occurs. Therefore, in our research, we applied three different sets of P/D programming pulses where the amplitudes were 0.6, 0.8, 1 V. The trade-off relationship between the on/off ratio and nonlinearity factor is

clearly demonstrated in a-Si CBRAM devices as shown in below Response Fig. 7 which is added as Supplementary Fig. 6 in the revised manuscript. However, our cluster-type Ti_{4.8%}:a-Si memristive device maintained an exceptionally low nonlinearity factor at 1 V pulse trains and even at 1.1 V pulse trains with the on/off ratio of 244.5 and nonlinearity factors of $\nu_P = 2.28$ and $\nu_D = 2.46$ (Fig. 2a). This result indicates that cluster-type switching methodology can achieve a superior analogue characteristic of both high on/off ratio and linear P/D at the same curve. Following your suggestion, we also inserted the description of the trade-off trend resulting from various pulse amplitude conditions. Thank you again for mentioning this critical issue that can support the superiority of our cluster-type memristive device.

Response Fig. 7 Average nonlinearity factor and on/off ratio of ten-P/D-cycle under three different pulse amplitudes. a a-Si (pristine) device. **b** a-Si (densified) device. **c** Ti_{4.8%}:a-Si device.

Changes in Manuscript

- In the Supplementary Information of revised manuscript, Response Fig. 7 is included as Supplementary Fig. 6 along with figure caption and additional information.
- In the revised manuscript page 6, the relevant description of this issue is added as **“These mechanisms of cluster-type analogue switching certainly lessened the positive feedback effect of an electric field and resolved the chronic trade-off problem (linearity and on/off ratio) of most filament-based CBRAMs as shown in Supplementary Fig. 6, maintaining an exceptionally low nonlinearity factor at all pulse amplitudes.”**

Comment #2

The authors refer to DC measurements; were these actually DC experiments or quasi-

static measurements.

Response

We would like to thank you for pointing out this key issue that might have hindered the readers understanding the resistive switching measurements of our devices. As the reviewer has pointed out, we misused the term “DC” for the quasi-static current-voltage (I - V) measurement that we have carried out in this research. It is our mistake for not clearly using the terminology, so following the reviewer’s comment, we removed the term “DC” and inserted “quasi-static,” throughout the manuscript. The specific voltage input condition we used in Keithley 4200A-SCS for the quasi-static I - V measurement is shown in the Response Fig. 8.

Response Fig. 8 An example of quasi-static current-voltage (I - V) measurement, voltage linear sweep operation mode (0 V \rightarrow +1 V \rightarrow 0 V) using the source measurement unit (SMU).

Changes in Manuscript

- In the revised manuscript, including Main text page 6, Figure caption 2b, Methods, and Supplementary Information, the terms “quasi-static” are added, and the terms “DC” are all removed.

Comment #3

Were the realized memristors characterized with respect to the elementary properties mentioned by Leon Chua?

Response

We show our regret for the use of confusing expressions and thank you for bringing out this fundamental issue. To our knowledge, the elementary memristor property mentioned by Leon Chua in 1971¹ has a functional relationship between charge and flux with memristance

M , $d\phi = Mdq$ (memristance M , charge q , and flux ϕ). However, our realized memristor is rather a memristive device or memristive system which is a much broader concept of memristor which was subsequently developed by Chua and Kang and has been widely used to address resistive switching effects. Accordingly, in our two-terminal nanoscale device, flux is no longer uniquely defined by the charge but described by the following two equations^{24,25}, $i = R(w, v)v$ and $dw/dt = f(w, v)$. The former equation is the I - V equation determined by instantaneous input (v) with the internal state variable (w). The state variable is further governed by the latter equation, also determined by instantaneous input and current state. Therefore, the device state can be acquired by time integral and exhibit history-dependent behavior. In our device, by the external voltage bias, the actual physical processes of the resistive switching occur during the set/reset operations based on coupled electronic and ionic transport mechanisms. The I - V characteristics of our fabricated devices are shown in Response Fig. 9 and they can be sorted as memristive devices. Thank you again for sharing your view and we are pleased to notify our updated sentences and descriptions in the revised manuscript, all of which were revised accordingly to your comment. Especially, we brought up the term “memristive device,” both in the Abstract and Introduction of the revised manuscript to elucidate our memristor device encompasses a broader concept of memristor.

Response Fig. 9 Quasi-static current-voltage (I - V) switching characteristics of the forming process with the following set and reset processes. **a** a-Si (pristine) device. **b** a-Si (densified) device. **c** and Ti_{4.8%}:a-Si device.

Changes in Manuscript

- In the Abstract of the revised manuscript page 1, the sentence is corrected as **“Memristors or memristive devices have ...”**
- In the revised manuscript page 2, additional information is added as **“The crossbar-**

structured memristors or memristive devices (throughout the paper, we use the term, “memristor,” in referring to a memristive device for short) in Fig. 1a.”

Comment #4

Is further linearization possible, physical limits?

Response

Thank you for referring the future work and limitations of our research. We have deeply ruminated over the issue and devised several ideas. Our cluster-type $\text{Ti}_{4.8\%}\text{-a-Si}$ memristive device demonstrated a linear potentiation/depression curve with its linearity value close to zero, which is the ideal linearity. In general filament-type CBRAM devices, near the maximum conductance level, it is difficult for a given pulse to increase the conductance, but it is easy to decrease the conductance²⁶. This phenomenon is mainly due to the positive feedback effect of an electric field during the filament growth process^{6,24}. In this respect, to implement the ideal linearity, there has to be no positive feedback effect at all, and thus the amount of conductance change must be same in any conductance state. Recently, there has been an attempt to improve the linearity by applying different type of pulse (e.g. pulse amplitude or width) depending on the current conductance of the device to minimize positive feedback effect of electric field (e.g. incremental step pulse programming). However, in this case, the current state of the device must be known in advance before the pulse is applied. Therefore, our cluster-type methodology is meaningful in that the linearity is greatly improved in the same pulse potentiation/depression measurement, sustaining large on/off ratio, by decreasing the positive feedback effect of electric field through Ag-Ti silicide clusters. We further extended our theoretical concepts of realizing the ideal linearity as below.

First, in terms of homogeneously dispersing the metal (Ag or Ti) clusters, the more uniformly distributed Ti silicide clusters will impact the linearity during the analogue measurement. In our research, the Ti atoms are incorporated in Si through RF co-sputtering method and if the Ti atoms can be more uniformly distributed by other doping procedure, the migrating Ag cations can more evenly reduced during the set/reset processes. In the same respect, an environment where Ag-Ti clusters grow evenly and prevent any particular cluster from enlarging abnormally, may contribute to demonstrating further linearization.

Second, regarding the material properties of the incorporated metals, a better linearity

may be obtained by using a material with a lower reduction potential. According to our research, the analogue linearity was selectively adjustable through the reduction potential of metals which applied to Ti, Ta, W, and Pt. However, we considered not only the reduction potential of the metals but also the thermodynamic miscibility with the switching medium, a-Si, and the active metal, Ag. Therefore, in our engineering methodology, Ti was optimal in enhancing analogue characteristics, but other switching and active material combinations may be more effective in linearity. Along with this concept, if a larger amount of reducing agents is inserted into the switching medium, the analogue linearity may be improved. In our situation, since the inserted Ti atoms are metal, a larger amount of Ti can greatly increase the initial conductance and significantly deteriorate the dynamic range. Therefore, a substance that does not affect the initial conductance state while acting as a reducing agent in the same way as Ti, may further linearize the analogue switching curve.

Changes in Manuscript

Comment #5

Please mention also device variations for these memristors.

Response

Thank you very much for pointing out an important device characteristic that we have missed to provide in the first manuscript submission. As you have suggested, we have evaluated the device-to-device variation of our cluster-type $\text{Ti}_{4.8\%}:\text{a-Si}$ device at ± 1 V, 1 μs pulse condition (0.1 V read pulse, 1 μs). The analogue conductance update curves of 20 devices are plotted by mean and mean \pm standard deviation (s.d.) in Response Fig. 10. Since the conductance level is not yet saturated, the size of the conductance update is the largest and accordingly, the s.d. values are larger in the earlier pulse number both in potentiation and depression. Following your comment, we have added the results as Supplementary Fig. 7 and the relevant description in the revised manuscript page 7, after the cycle-to-cycle variation in Supplementary Fig. 4. Thank you again for your valuable comment.

Response Fig. 10 Device-to-device variation of the Ti_{4.8%}:a-Si memristor. Analogue conductance update curves of 20 devices are plotted by mean and mean \pm standard deviation (s.d.).

Changes in Manuscript

- In the revised manuscript page 7, the description of Supplementary Fig. 7 is added as **“... and device-to-device variation along with standard deviation (s.d.) at each P/D pulse number (Supplementary Fig. 7).”**
- In the Supplementary Information of revised manuscript, Response Fig. 10 is included as Supplementary Fig. 7 along with the figure caption and additional information.

Comment #6

Page 2: The memristor was not rediscovered in 2008 but experimental results of a fabricated device have been related to the continuously developed theory of Leon Chua starting in 1971.

Response

We are very grateful for bringing out this important issue to us. As you have pointed out, the memristor was not rediscovered in 2008 but rather the electronic device of two-terminal metal-insulator-metal (MIM) structure memristive system have been developed, which is closely related to the memristor theory devised by Leon Chua in 1971¹. We also should have stated the term “memristive device or memristive system” with clarity in the beginning of our paper, so as not to puzzle the readers about the specific electronic device we fabricated. We again appreciate your insightful remark on this matter.

Changes in Manuscript

- In the revised manuscript page 2, the sentence is rewritten as **“After the experimental demonstration of the two-terminal metal-insulator-metal (MIM) structure memristive system in 2008¹, the nanoscale ...”**
- In the Abstract of the revised manuscript page 1, the sentence is corrected as **“Memristors or memristive devices have ...”**
- Moreover, in the revised manuscript page 2, additional information is added as **“The crossbar-structured memristors or memristive devices (throughout the paper, we use the term, “memristor,” in referring a memristive device for short) in Fig. 1a.”**

Comment #7

Page 6: The authors refer to an optimization that is usually done mathematically but here they are referring to a material selection with different values of E^ϕ values. This point may be better clarified.

Response

We highly appreciate your critical and thoughtful suggestion. As you have pointed out, we fully agree to our not clarifying the expression “optimization or optimized” which meant a methodology that realizes maximized analogue linearity in cluster-type memristor by adding transition-metal reducing agents. We also misreferred the term “optimized” in indicating the $Ti_{4.8\%}:a-Si$ device and the analogue property used in the feature extraction task simulation. Thus, in the revised manuscript, we removed and changed the terms for better clearance and readability. Thank you again for pointing out the ambiguity in our word selection.

Changes in Manuscript

- In the revised manuscript page 6 and 10, we removed the unnecessary term “optimized.”
- In the revised manuscript page 7, we rewrote the sentences as **“In terms of retention characteristic, our Ti-assisted device ...,”** and **“... further development of ...”**
- In the revised manuscript page 10, we updated the word as **“Ti:a-Si device case.”**
- In the revised manuscript page 11, we changed “optimized” to **“enhanced.”**

Comment #8

Page 10: The authors should explain and specify the “experimental model”

Response

We deeply regret our lack of explanation on the “experimental model” mentioned in the first submitted manuscript page 10. The expression we used in the first submitted manuscript meant the nonlinearity factor extracted from the actual experimental analogue data measured by potentiation/depression pulses. Accordingly, the model we employed was not analytical model but was to simply minimize the absolute value of the difference between the fitting curve and the experimental data at each pulse from the mathematical equations (equations in nonlinearity value calculations in Methods). Therefore, we removed the term “model” and instead articulated that the feature extraction tasks are based on the experimental nonlinearity values extracted by the nonlinearity value calculation process at the beginning of the sparse coding simulation subheading. Along with these modifications, we updated the term “device model” to “device case or characteristic” for readers to clearly comprehend the simulation tasks in our research. Thank you again for your wonderful remark.

Changes in Manuscript

- In the revised manuscript page 10, we rewrote the subheading and the following sentences as **“Feature extraction task using sparse coding with the experimentally measured analogue characteristics of memristor device.”** and **“Based on the experimental nonlinearity values extracted by the nonlinearity value calculation process (see Methods for details) ...”**
- In the revised manuscript, we updated the term “device model” to **“device case or characteristic”** in Fig. 5e, Supplementary Fig. 18,19, page 3, 10, and 11.
- In the revised manuscript page 19, we added the sentence **“The nonlinearity factors for potentiation (ν_p) and depression (ν_d) are calculated by fitting the above equations through minimising the absolute difference between the fitting curve and the experimental results at every pulse.”**

Comment #9

Page 10 The sparse coding applications – including settings and parameters - should be described more detailed.

Response

Thank you for pointing out our lack of details describing the initial settings and parameters of sparse coding application. To elaborate the details, we first added the actual values of the pre-defined parameters in the main text and methods section, the learning rate ($\beta = 0.0004$) and time constant ($\tau = 0.008$) which are remained constant in all simulation tasks. The pixel patch information is also included in the text and Fig. 5b caption in the revised manuscript. Second, we updated the sentences that with these fixed parameters (β and τ), we compared the sparsity and mean squared error results for various threshold (λ) values. Moreover, the optimal result condition of the sparse coding application is stated for the readers to better understand this simulation results. Lastly, in the methods section of the revised manuscript, we included the MSE calculation equations between the original image pixels and the reconstructed image pixels. Your suggestion has been a great help improving the delicacy of the sparse coding application details.

Changes in Manuscript

- In the revised manuscript page 11, the additional information of the initial simulation settings and parameters are added as **“Throughout the receptive field learning by SGD, a learning rate parameter (β) value of 0.0004 is used.”** and **“During this task, the 140×140 resolution natural image is broken into 20×20 pixel patches each. The time constant parameter (τ) value of 0.008 is used during the feature extraction task by LCA.”**
- In the revised manuscript page 11, the descriptions of the sparse coding application are included as **“With these parameter settings, we compared the sparsity (L_0 -norm, number of active neurons) and mean squared error (MSE) results for various threshold (λ) values. The optimal result of the sparse coding application is conditioned by high sparsity (low L_0 -norm) and low MSE ... MSE results are summarised in Fig. 5d, where the Ti_{4.8%}:a-Si device case successfully demonstrated much higher reconstruction accuracy than that of the a-Si (pristine) device case, close to an ideal case. The reconstructed input images using the sparse codes and feature matrix are shown in Fig. 5e, where the Ti_{4.8%}:a-Si device case can reproduce comparable L_0 -norm and MSE images to the ideal case at the identical λ value.”**

- In the Methods of the revised manuscript page 21-22, the information is added as “... β is the learning rate ($\beta = 0.0004$ is used).” and “... time constant τ ($\tau = 0.008$ is used).”
- In the Methods of the revised manuscript page 22, the descriptions of the MSE calculation and equation are added as “The mean squared error (MSE) is calculated with the following equation⁵⁵:

$$MSE = \frac{1}{400} \sum_{i=1}^{400} (X - a \cdot \Phi^T)_i^2$$

Comment #10

Fig. 1a: The blue graphics are confusing.

Response

Thank you for your keen observation, which made us realize that the biological synapse illustration is too transparent and also ambiguous in explaining which part of the electronic device it resembles. Accordingly, we made the graphic non-transparent, improved its clarity, and enlarged the important part, synapse. Moreover, we altered the arrangement of SEM image and synapse graphic for the readers’ better understanding.

Changes in Manuscript

- In the Fig. 1a of the revised manuscript, the Fig. 1a is changed as below.

- In the Fig. 1a caption of the revised manuscript, the sentence is rewritten as “left inset: scanning electron microscopy (SEM) image of crossbar structure, right inset: illustrated biological synapse.”

Comment #11

Fig. 1b-d: Please describe the dynamics in the figure caption.

Response

Thank you for raising the missing details regarding the description of switching dynamics in Fig. 1b-d caption. Adhering to your advice, we have included the information of the switching dynamics corresponding to each figure (b to d): **b** high cation mobility (μ) case for a-Si (pristine), **c** low μ case for a-Si (densified), and **d** low μ and high Ag reduction probability ($\Gamma_{\text{Ag}}^{\text{red}}$) case for Ti_{4.8%}:a-Si device.

Changes in Manuscript

- In the Fig. 1b-d caption of the revised manuscript page 13, the sentence is updated as **“Device schematics with operation mechanism illustration for each memristor situation, b high cation mobility (μ) case for a-Si (pristine) device, c low μ case for a-Si (densified) device, and d low μ and high Ag reduction probability ($\Gamma_{\text{Ag}}^{\text{red}}$) case for Ti_{4.8%}:a-Si device.”**

Comment #12

Fig. 2b: The underlying curve might look rather nonlinear?

Response

Thank you for your valuable comment regarding the I - V characteristic of Ti_{4.8%}:a-Si device in Fig. 2b (Response Fig. 11b). We are sorry for making confusion by using the term “gradual” in the first submitted manuscript page 6 to describe the I - V switching characteristic. Indeed, from the I - V switching measurement, we cannot correctly represent the analogue performance of the memristive device, which is why we should have avoided the term “gradual” and specify that the linear characteristic is obtained from analogue measurement (pulse measurement). However, the analogue properties can be partially anticipated through I - V measurement. In case of our device, the a-Si (pristine) device showed abrupt set/reset processes I - V curve, resulting highly digital (nonlinear) switching in analogue measurement (Response Fig. 11a). The Ti_{4.8%}:a-Si device exhibited non-abrupt set/reset processes in I - V curve, exhibiting highly analogue (linear) switching in analogue measurement (Response Fig. 11b). Therefore reflecting your comment, we removed the word “gradual” in referring I - V curve in Fig. 2b, and added relevant information about the abrupt and non-abrupt I - V set/reset processes occurring in a-Si (pristine) and Ti_{4.8%}:a-Si devices, respectively.

Response Fig. 11 The quasi-static I - V switching characteristic (red and blue) and analogue conductance update measurement (black) in **a** a-Si (pristine) device and **b** Ti_{4.8%}:a-Si device.

Changes in Manuscript

- In the revised manuscript page 7, we removed the term “gradual.”
- In the revised manuscript page 7, we added the sentence **“Interestingly, the I - V set/reset processes in the Ti_{4.8%}:a-Si device contrasted with much more abrupt I - V set/reset processes occurring in the a-Si (pristine) device (Supplementary Fig. 1a).”**

Comment #13

Page 18: G_1 is not defined

Response

We would like to apologize for the absence of the G_1 's definition and thank the reviewer for pointing out this matter. G_1 is the function of parameter, nonlinearity value (ν), to suit the G_P and G_D functions within the range of G_{\max} , G_{\min} , and N_{\max} . For a better understanding of nonlinearity extracting equations, we have added not only the description of G_1 but also specified the equations in detail as “ $1 - N$ ” to “ $N_{\max} - N$,” and “ $-\nu$ ” to “ $-\nu N_{\max}$,” respectively. This revision is because the 1 in the equation of the previously submitted manuscript meant

maximum normalized pulse number, N_{\max} .

Changes in Manuscript

- In the Methods of the revised manuscript page 19, we have added descriptions about N_{\max} and G_1 as “ N and N_{\max} are the normalised pulse number and maximum normalised pulse number, respectively, the latter of which is 1. G_1 is the function of ν in order to fit (normalise) the G_P and G_D functions within the range of G_{\max} , G_{\min} , and N_{\max} .”
- At the same section, the mathematic equations are rewritten as below.

$$G_P = G_1(1 - e^{-\nu_P N}) + G_{\min}$$
$$G_D = G_{\max} - G_1[1 - e^{-\nu_D(N_{\max} - N)}]$$
$$G_1 = \frac{G_{\max} - G_{\min}}{1 - e^{-\nu N_{\max}}}$$

Comment #14

Define all quantities used in the manuscript, also when known in the community of the authors, e.g. E° .

Response

Thank you very much for your help, which made us realize the missing definition of the important quantities. In the first submitted manuscript, we have mentioned the quantity E^ϕ as the standard reduction potential of various metals. However, when we referred to the actual values of the reduction potentials, we wrote the units as “ E°/V ,” which is incorrectly and should be changed to simply “V.” Since both E^ϕ and E° are the same, meaning the standard reduction potential, we removed all the misused quantities, “ E°/V ,” and changed simply to “V” to indicate the unit of all standard reduction potential of metals (e.g. -1.63 E°/V to -1.63 V). Moreover, we updated the missing definitions, such as G_1 and N_{\max} in the revised manuscript page 19 (comment #13) and E in the revised manuscript page 21.

Changes in Manuscript

- In the main text, including in Fig. 1d and 4a, we modified the unit “ E°/V ,” to “V.”
- In the Methods of the revised manuscript page 21, we have added the description of E

as “*E is the error with respect to the dictionary ...*”

Comment #15

Finally, the paper should be checked by a native speaker for improvement.

Response

Thank you for you commenting on the mistakes regarding our language usage in the first submitted manuscript. Thanks to your thoughtful suggestions, we have made minor revisions to the sentence structures and words. Additionally, we had our manuscript checked by a native speaker with hopes of improving its readability. Along with these changes, we have also modified the manuscript’s format following the instructions of *Nature Communications*. Finally, we have adhered to the British English writing style throughout the paper (e.g. characterisation, mamixised, and summarised). We again thank for the reviewer’s kind comment.

References

1. Chua, L. O. *Memristor-The Missing Circuit Element*. *EEE TRANSACTIONS ON CIRCUIT THEORY* vol. 18 (1971).
2. Zhu, J., Zhang, T., Yang, Y. & Huang, R. A comprehensive review on emerging artificial neuromorphic devices. *Applied Physics Reviews* **7**, (2020).
3. Yang, Y. *et al.* Observation of conducting filament growth in nanoscale resistive memories. *Nature Communications* **3**, 732–738 (2012).
4. Yang, Y. *et al.* Electrochemical dynamics of nanoscale metallic inclusions in dielectrics. *Nature Communications* **5**, (2014).
5. Tsuruoka, T. *et al.* Redox Reactions at Cu,Ag/Ta2O5 Interfaces and the Effects of Ta2O5 Film Density on the Forming Process in Atomic Switch Structures. *Advanced Functional Materials* **25**, 6374–6381 (2015).
6. Valov, I., Waser, R., Jameson, J. R. & Kozicki, M. N. Electrochemical metallization memories - Fundamentals, applications, prospects. *Nanotechnology* **22**, (2011).
7. Jeong, Y. J., Kim, S. & Lu, W. D. Utilizing multiple state variables to improve the dynamic range of analog switching in a memristor. *Applied Physics Letters* **107**, (2015).
8. Yeon, H. *et al.* Alloying conducting channels for reliable neuromorphic computing. *Nature Nanotechnology* **15**, 574–579 (2020).
9. Kim, K. *et al.* Enhanced analog synaptic behavior of SiNx/a-Si bilayer memristors through Ge implantation. *NPG Asia Materials* **12**, (2020).
10. Choi, S. *et al.* SiGe epitaxial memory for neuromorphic computing with reproducible high performance based on engineered dislocations. *Nature Materials* **17**, 335–340 (2018).
11. Zhao, H. *et al.* Atomically Thin Femtojoule Memristive Device. *Advanced Materials* **29**, 1–7 (2017).
12. Sheridan, P. M., Du, C. & Lu, W. D. Feature Extraction Using Memristor Networks. *IEEE Transactions on Neural Networks and Learning Systems* **27**, 2327–2336 (2016).
13. Sheridan, P. M. *et al.* Sparse coding with memristor networks. *Nature Nanotechnology* **12**, 784–789 (2017).
14. Jeong, Y., Zidan, M. A. & Lu, W. D. Parasitic Effect Analysis in Memristor-Array-Based Neuromorphic Systems. *IEEE Transactions on Nanotechnology* **17**, 184–193 (2018).
15. Wang, Z. *et al.* Memristors with diffusive dynamics as synaptic emulators for neuromorphic computing. *Nature Materials* **16**, 101–108 (2017).
16. Sebastian, A., le Gallo, M., Khaddam-Aljameh, R. & Eleftheriou, E. Memory devices and applications for in-memory computing. *Nature Nanotechnology* **15**, 529–544 (2020).

17. Zhang, W. *et al.* Neuro-inspired computing chips. *Nature Electronics* **3**, 371–382 (2020).
18. Yin, S. *et al.* Emulation of Learning and Memory Behaviors by Memristor Based on Ag Migration on 2D MoS₂ Surface. *Physica Status Solidi (A) Applications and Materials Science* **216**, 1–8 (2019).
19. Gonzalez-Rosillo, J. C. *et al.* Lithium-Battery Anode Gains Additional Functionality for Neuromorphic Computing through Metal–Insulator Phase Separation. *Advanced Materials* **32**, 1–12 (2020).
20. Ambrosi, E., Bricalli, A., Laudato, M. & Ielmini, D. Impact of oxide and electrode materials on the switching characteristics of oxide ReRAM devices. *Faraday Discussions* **213**, 87–98 (2019).
21. Tsuruoka, T., Hasegawa, T., Valov, I., Waser, R. & Aono, M. Rate-limiting processes in the fast SET operation of a gapless-type Cu-Ta₂O₅ atomic switch. *AIP Advances* **3**, 1–7 (2013).
22. Lenser, C. *et al.* Insights into nanoscale electrochemical reduction in a memristive oxide: The role of three-phase boundaries. *Advanced Functional Materials* **24**, 4466–4472 (2014).
23. Wong, H. S. P. *et al.* Phase change memory. in *Proceedings of the IEEE* vol. 98 2201–2227 (Institute of Electrical and Electronics Engineers Inc., 2010).
24. Yang, Y. & Lu, W. Nanoscale resistive switching devices: Mechanisms and modeling. *Nanoscale* **5**, 10076–10092 (2013).
25. Strukov, D. B., Snider, G. S., Stewart, D. R. & Williams, R. S. The missing memristor found. *Nature* **453**, 80–83 (2008).
26. Agarwal, S. *et al.* Resistive memory device requirements for a neural algorithm accelerator. in *2016 International Joint Conference on Neural Networks (IJCNN)* 929–938 (IEEE, 2016).

REVIEWERS' COMMENTS

Reviewer #2 (Remarks to the Author):

The authors have fully addressed my concerns. This work is not ready for publication. I would like to suggest the authors to polish further for more comprehensive study in regards to alloy strategies in memristors.

Reviewer #3 (Remarks to the Author):

Many thanks for taking care for my points with regard to your answer to comment 3, I would like to mention that you may study

If it's pinched it's a memristor

Leon Chua

Published 18 September 2014 • © 2014 IOP Publishing Ltd
Semiconductor Science and Technology, Volume 29, Number 10 Citation Leon Chua 2014 Semicond. Sci. Technol. 29 104001

since the term extended memristor may be appropriate as it will be used in most of the publications.
Finally do not use "dictionary" on page 21.

AUTHORS' RESPONSE TO THE REFEREES' COMMENTS

We greatly appreciate the editor and all the referees for generously lending us help and sharing their valuable insights on our research. We are especially grateful for spending your valuable time on reviewing our manuscript and providing scientific insights, which greatly improved the quality of our manuscript. Thanks to all of your dedication, we have made revisions for the final submission of our manuscript. In the following letter, we provide point-by-point responses to the referees' comments. The changes are indicated as an **orange color** in both revised manuscript and response letter.

Reviewer #2

General comment

The authors have fully addressed my concerns. This work is not ready for publication. I would like to suggest the authors to polish further for more comprehensive study in regards to alloy strategies in memristors.

Response

We are glad that the reviewer found the revision satisfactory, and we express our sincere appreciation for your valuable dedication on our manuscript. Indeed, we feel the need to delineate the alloy strategy guidelines more robustly to help improve analogue properties in CBRAMs as well as in various types memristive devices. Therefore, as the reviewer suggested, we will proceed to conduct the additional research regarding alloy formation mechanisms and hope that we present the results to the researchers in the near future. Thank you again for contributing to the improvement of our research and spending your time on reviewing our manuscript.

Reviewer #3

General comment

Many thanks for taking care for my points with regard to your answer to comment 3, I would like to mention that you may study.

If it's pinched it's a memristor

Leon Chua

Published 18 September 2014 • © 2014 IOP Publishing Ltd

Semiconductor Science and Technology, Volume 29, Number 10 Citation Leon Chua 2014
Semicond. Sci. Technol. 29 104001

since the term extended memristor may be appropriate as it will be used in most of the publications. Finally do not use "dictionary" on page 21.

Response

We sincerely appreciate your constructive comments that considerably helped to raise the quality of our work. Your great suggestion has been an excellent help improving our paper's details. Also, we will refer to the recommended Chua's paper and make sure to utilize to it in the following study. Once again, thank you for your participation as a reviewer of our study and we wish you all the best in your future endeavors.

Changes in Manuscript

- In the revised manuscript, figure legend, and Supplementary Information, we have changed the term "dictionary" as "**receptive fields.**"